# Nuclear export inhibition jumbles epithelial–mesenchymal states and gives rise to migratory disorder in healthy epithelia

**Carly M Krull[1], Haiyi Li[2], Amit Pathak[1,3]\***

[1]Department of Biomedical Engineering, Washington University in St. Louis, St Louis, United States; [2]Department of Computer Science and Engineering, Washington University in St. Louis, St Louis, United States; [3]Department of Mechanical Engineering and Materials Science, Washington University in St. Louis, St Louis, United States

**Abstract** Dynamic nucleocytoplasmic transport of E-M factors regulates cellular E-M states; yet, it remains unknown how simultaneously trapping these factors affects epithelia at the macroscale. To explore this question, we performed nuclear export inhibition (NEI) via leptomycin B and Selinexor treatment, which biases nuclear localization of CRM1-associated E-M factors. We examined changes in collective cellular phenotypes across a range of substrate stiffnesses. Following NEI, soft substrates elevate collective migration of MCF10A cells for up to 24 hr, while stiffer substrates reduce migration at all time points. Our results suggest that NEI disrupts migration through competition between intercellular adhesions and mechanoactivation, generally causing loss of cell–cell coordination. Specifically, across substrate stiffnesses, NEI fosters an atypical E-M state wherein MCF10A cells become both more epithelial and more mesenchymal. We observe that NEI fosters a range of these concurrent phenotypes, from more epithelial shYAP MCF10A cells to more mesenchymal MDCK II cells. α-Catenin emerges as a potential link between E-M states, where it maintains normal levels of intercellular adhesion and transmits mechanoactive characteristics to collective behavior. Ultimately, to accommodate the concurrent states observed here, we propose an expanded E-M model, which may help further understand fundamental biological phenomena and inform pathological treatments.

**\*For correspondence:**
pathaka@wustl.edu

**Competing interest:** The authors declare that no competing interests exist.

## Editor's evaluation

This work is an important contribution to our understanding of epithelial migration. Previous work had shown that nuclear export inhibition (NEI), which is employed as a therapeutic strategy to treat cancer, traps several known regulators of epithelial-mesenchymal (E-M) phenotypes; however, how NEI alters the mechano-response and collective cell migration of healthy epithelia on substrates of varying stiffness was not described. The convincing new results show that NEI induces an intermediate E-M state where cells concurrently strengthen intercellular adhesions and develop mechano-active characteristics. Migration of epithelial monolayers becomes disordered and leads to multicellular streaming.

## Introduction

Collective cell migration facilitates diverse tissue functions, ranging from normal physiological activities, like embryogenesis and tissue repair, to malignant processes, like cancer invasion. While carrying out these functions, grouped cells can manifest diversely – as monolayers, clusters, or streams

(*Etienne-Manneville, 2014*). Regardless of appearance, coordinated migration of the group is a carefully orchestrated balance between intercellular adhesive forces and intracellular propulsive ones (*Trepat and Sahai, 2018*). Cells transmit active forces from myosin molecular motors to their substrate (i.e., traction) to enable their forward motion. Meanwhile, those same motors generate tensile stress across adherens junctions, imparting strength to intercellular adhesions (*Alert and Trepat, 2020*). Cellular phenotypes arise, in part, from the degree of balance between these forces. When intercellular adhesion is high and anisotropic intracellular force is low, cells appear epithelial, presenting with apicobasal polarization and minimal motility (*Yang et al., 2020*). Reversal of that force balance gives rise to a mesenchymal phenotype; cells become more motile and invasive, polarizing front–back in their direction of migration. This shift in cellular characteristics constitutes a process termed epithelial–mesenchymal transition (EMT). However, far from a phenotypic switch, the transition between epithelial and mesenchymal states encompasses a spectrum of intermediate phenotypes, where a range of moderate intercellular adhesion and substrate tractions coincide. Ultimately, it is these intermediate states that enable the varying modes of collective cell migration.

Migrating epithelia comprise distinct leader and follower cell populations, which cooperate to facilitate group motion (*Desai et al., 2013*; *Qin et al., 2021*). Cells at the edge, the leaders, exhibit higher polarity and generate higher tractions (*Reffay et al., 2011*; *Reffay et al., 2014*; *Trepat and Sahai, 2018*). These cells apply most of the force required for locomotion, and they retain lower intercellular adhesion to do so (*Matsuzawa et al., 2018*). Meanwhile, cells behind – the followers – supplement leader tractions. Whereas followers maintain better adhesion to neighbors, they also extend cryptic protrusions to help forwardly propel the collective (*Qin et al., 2021*; *Trepat and Sahai, 2018*). What arises from these distinct cell populations is an EMT gradient that fundamentally supports migration (*Sarker et al., 2019*). This gradient transpires from the tactful regulation of transcription factors (TFs) and migration-related proteins. Core EMT TFs, including Zeb, Snail, Slug, and Twist directly regulate E-M phenotypes *Yang et al., 2020*; and auxiliary migration-related proteins (e.g., YAP, IκBα, SOX9, and HIF2A, FOXA2) further contribute to movement between E-M states (*Park et al., 2019*; *Huber et al., 2004*; *Huang et al., 2019*; *Yang et al., 2016*; *Zhang et al., 2015*). However, while, on their own, these proteins unidirectionally affect cell fate (i.e., promote or inhibit EMT), it is unknown whether opposing factors compete, cooperate, or cancel during the construction of cell phenotypes.

To explore this question, we consider that the ability of EMT-TFs and related proteins to engender transcriptional changes hinges on their location within the cell: nuclear localization facilitates transcription of target genes, and cytoplasmic sequestration prevents it (*Köster et al., 2005*). Such trafficking between the nucleus and cytoplasm is termed nucleocytoplasmic (NC) transport (*Cartwright and Helin, 2000*). It denotes a highly controlled process, where nuclear import and export receptors ferry protein cargos through the nuclear pore complex to coordinate the signaling required for

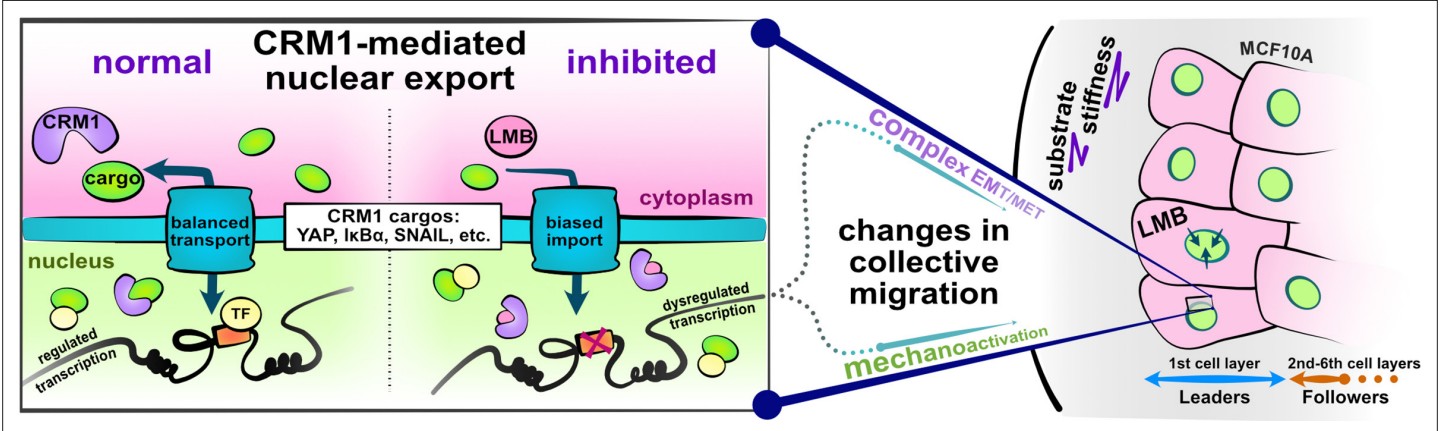

**Figure 1.** Schematic depicting experimental design and study hypotheses. The enlarged panel illustrates normal nucleocytoplasmic transport and accompanying well-regulated gene transcription (left). Changes following CRM1 inhibition (right) enable examination of the interaction between opposing epithelial–mesenchymal transition (EMT)-related proteins in the development of collective cell phenotypes and migration characteristics. Depicted cells denote the experimental setup: MCF10A epithelial monolayers on polyacrylamide gels of varying stiffness, where leptomycin B is used to inhibit CRM1-mediated nuclear export.

proper cell function. One way to survey interactions between opposing EMT factors is to force the nuclear co-localization of EMT-related proteins by biasing their import. Interfering with the appropriate nuclear export receptor achieves this outcome, and this process is called nuclear export inhibition (NEI). NEI has already been developed as a therapeutic strategy for cancer, where inhibition of the nuclear export receptor CRM1 returns critical tumor suppressor and oncogenic proteins to the nucleus and helps restore cell cycle regulation (*Gravina et al., 2014*; *Gravina et al., 2017*). However, CRM1-based NEI also biases the nuclear import of several key EMT promoters (i.e., SNAIL, YAP, SOX9, HIF2A) and inhibitors (i.e., IκBα, FOXA2) (*Xu et al., 2012*). Here, we leverage this function of CRM1-based inhibitors to determine how nuclear co-localization of opposing EMT-related proteins manifests phenotypically. Interestingly, previous studies have found evidence only that CRM1-based NEI reverses EMT (*Gravina et al., 2014*; *Azmi et al., 2015*; *Gravina et al., 2017*; *Kashyap et al., 2016a*; *Kashyap et al., 2016b*; *Galinski et al., 2021*). However, E-M characteristics during NEI have been investigated thus far in a relatively limited context, discussed in terms of one or a few markers. Therefore, we wondered whether cellular outcomes could be more intricate than initially credited. To test this hypothesis, we treat monolayers composed of healthy MCF10A human mammary epithelial cells, chosen for their known E-M plasticity and mechanosensitivity, with the CRM1-based inhibitor of nuclear export, leptomycin B (LMB) (*Kudo et al., 1999*). We then assess changes in E-M cellular features and collective migration characteristics (*Figure 1*). To assess whether the type of cell or method of induced NEI influence phenotypic outcomes, we also report findings for MDCK I and II cells, and for the selective inhibitor of nuclear export (SINE), Selinexor.

Because this experimental framework depends greatly on NC transport, we also consider that the movement of proteins through nuclear pores is highly dynamic. Thus, the relative rates of import and export decide transcriptional outcomes. These rates are set by cargo size and geometry but are subject to change depending on the stiffness of the underlying substrate (*Elosegui-Artola et al., 2017*). More precisely, high stiffness promotes nuclear flattening and increases the nuclear import of proteins with higher molecular weight and stability. This process is one of the ways that stiffness-induced cellular mechanoactivation occurs, that is, through Yes-associated protein 1 (YAP)-mediated signaling. Therefore, amidst competing EMT cues, substrate stiffness may contribute rate-dependent differences or additional mechanoactive features to phenotypic outcomes (*Walter et al., 2018*; *Fattet et al., 2020*; *Wei et al., 2015*). To uncover this potential stiffness dependency, we compare NEI effects for MCF10A epithelia seeded on polyacrylamide gels spanning a range of stiffnesses.

Given the complexity of EMT signaling arising from the combined NEI and substrate cues, in this study we describe epithelial and mesenchymal cellular traits comprehensively, using physical characteristics, migratory phenotypes, together with candidate gene and protein expressions, as discussed previously (*Yang et al., 2020*). Ultimately, we find that NEI generates – in parallel – cellular characteristics classically categorized as epithelial or mesenchymal, and ultimately forces competition between cellular E-M states. What arises is highly disordered collective migration; yet, whether NEI promotes or inhibits migration depends distinctly on the underlying matrix stiffness. Together, these experiments demonstrate that NEI traverses length scales – concurrently trapping E-M factors, jumbling cellular E-M states, and disrupting grouped epithelial migration. Overall, the presented work suggests that epithelial and mesenchymal states are not necessarily mutually exclusive, which has implications for how cells may manipulate phenotypes to promote cancer, wound healing, and embryonic development.

## Results
### NEI reinforces epithelial characteristics in a stiffness-dependent manner
Previous investigations have proposed that NEI reverses EMT, and mechanistic studies indicate this reversal may transpire via NFκB Inhibitor Alpha (IκBα)-dependent inhibition of Nuclear Factor Kappa B (NFκB) (*Figure 2A*; *Gravina et al., 2014*; *Azmi et al., 2015*; *Gravina et al., 2017*; *Kashyap et al., 2016a*; *Kashyap et al., 2016b*; *Galinski et al., 2021*). However, E-M characteristics during NEI have been investigated thus far in a relatively limited context, discussed in terms of one or a few markers, and with culture performed primarily on tissue culture plastic. Because NEI causes nuclear localization of multiple competing EMT-related proteins and stiff substrates themselves trigger EMT through mechanoactive signaling, we wondered whether the manner in which NEI changes cellular E-M state

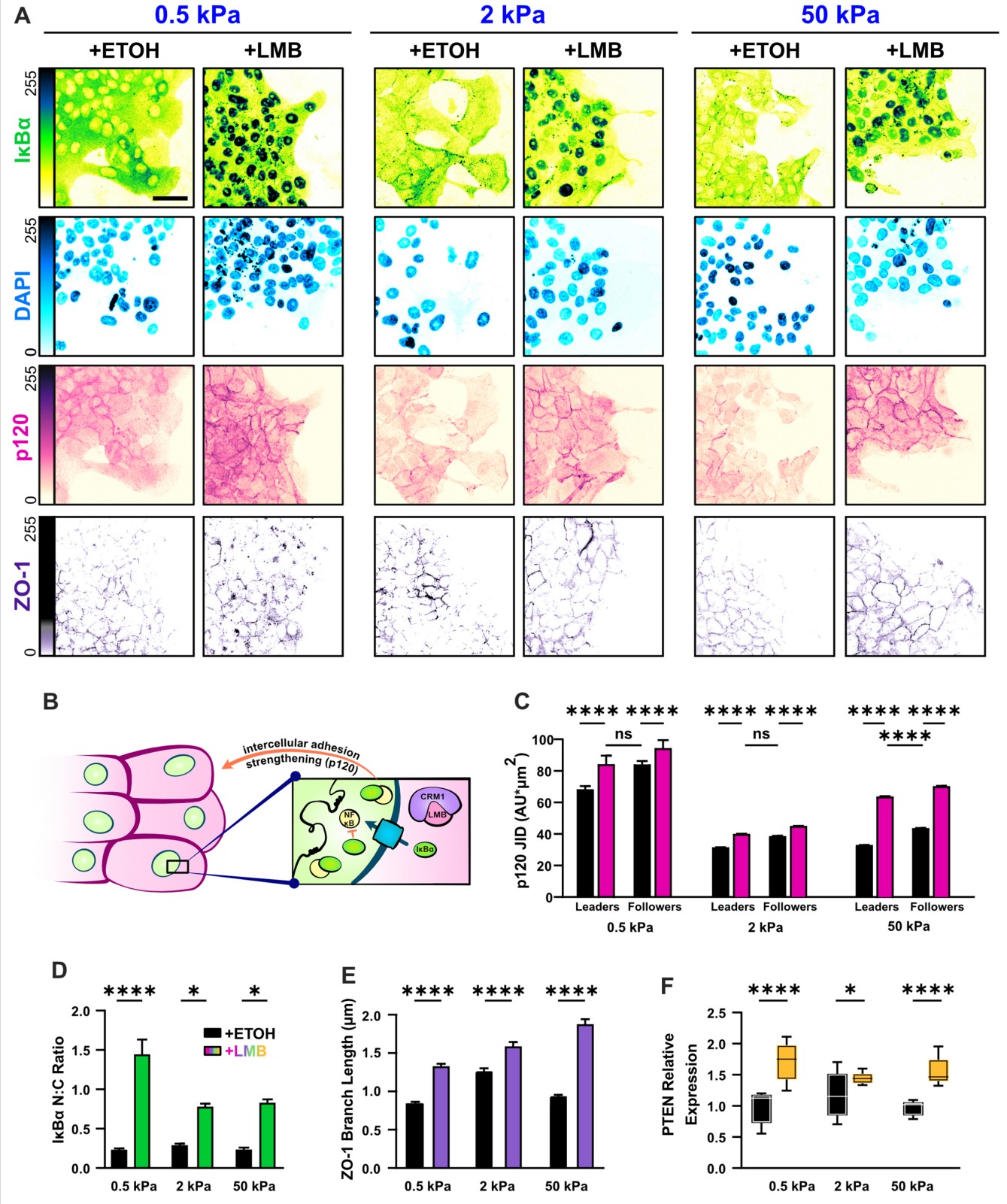

**Figure 2.** Nuclear export inhibition (NEI) enriches epithelial features in MCF10A collectives. (**A**) Representative images for epithelial characteristics of WT MCF10A on 0.5, 2, and 50 kPa polyacrylamide gels. Monolayers were treated with ethanol (ETOH) as vehicle or leptomycin B (LMB) for NEI. Images depict nucleocytoplasmic localization of IκBα (left), p120 expression (middle), and DAPI (4',6-diamidino-2-phenylindole) nuclear signal (right). (**B**) Schematic illustrating the established relationship between LMB and cell–cell adhesion strengthening, where nuclear accumulation of IκBα promotes

*Figure 2 continued on next page*

*Figure 2 continued*

inhibition of NF $\kappa$ B. (**C**) Leader–follower changes in p120 junction integrated density (JID) ($n > 39$ leaders and 55 followers). (**D**) Nucleocytoplasmic (N:C) ratio for I $\kappa$ B$\alpha$ ($n = 8$). (**E**) ZO-1 branch length ($n > 4500$). (**F**) Relative gene expression for epithelial marker PTEN ($n = 12$). Data were analyzed using a two-way analysis of variance (ANOVA) to evaluate NEI and stiffness effects. Data were analyzed using a three-way ANOVA with Tukey post hoc analyses to evaluate NEI, stiffness, and leader–follower differences. Significance levels: * < .0332, **** < 0.0001. Bars represent mean ± standard error of the mean (SEM). Scale bar: 50 μm.

could be more intricate than initially credited. To investigate this question, we perform a broad analysis of cell phenotype across mechanically varying substrates. Here, we begin by assessing I$\kappa$B$\alpha$ localization and intercellular adhesion strength to determine how NEI influences epithelial characteristics in our model system.

Nuclear localization of I$\kappa$B$\alpha$ is an established measure for NF$\kappa$B inhibition (***Arenzana-Seisdedos et al., 1997***), and junctional expression of Catenin Delta 1 (p120) reflects the degree of cell–cell adhesion reinforcement (***Kourtidis et al., 2013***; ***Figure 2B***). Therefore, to determine whether epithelial characteristics changed with substrate stiffness, we surveyed the localization of I$\kappa$B$\alpha$ and junctional p120 levels on 0.5, 2, and 50 kPa polyacrylamide gel. Consistent with prior studies in cancer cells, we found that NEI contributed to higher I$\kappa$B$\alpha$ nucleocytoplasmic (N:C) ratios (***Figure 2A, D***). The preferential nuclear localization of I$\kappa$B$\alpha$ developed in both leader and follower cells. However, the strength of epithelial characteristics did depend on substrate stiffness. Nuclear localization of I$\kappa$B$\alpha$ was higher on 0.5 kPa than on 2 and 50 kPa, and junctional p120 levels exhibited a matching stiffness effect (***Figure 2C***). Yet remarkably, across stiffnesses, the conventional relationship between leader and follower cells – where followers adhere more strongly to their neighbors – was preserved. Together, these findings suggest that mechanoactivation, as translated through substrate stiffness, limits NEI's ability to inhibit NF$\kappa$B and reinforce cell–cell adhesions. Despite this, NEI upholds traditional leader–follower cell–cell adhesion gradients independent of substrate stiffness.

Beyond this established mechanism of epithelial reinforcement during NEI, we wanted to determine whether other traditionally epithelial markers were affected. We measured expression of the epithelial marker gene Phosphatase And Tensin Homolog (PTEN), and found that it was upregulated across substrate stiffnesses (***Figure 2F***). This is consistent with previous studies that demonstrated an antagonistic interaction between NF$\kappa$B activation and PTEN expression (***Kim et al., 2004***). We then examined whether NEI changed the integrity of tight junctions by measuring contiguous length of Tight Junction Protein 1 (ZO-1). We found that NEI significantly increased ZO-1 branch length across substrate stiffness (***Figure 2D***). Altogether, these findings of elevated nucleocytoplasmic I$\kappa$B$\alpha$ ratio, increased protein expression of p120 and ZO-1, and upregulated PTEN gene expression indicate that NEI reinforces epithelial characteristics.

## Mechanoactivation during NEI evolves together with reinforced intercellular adhesion to promote a concurrent epithelial–mesenchymal state

Because NEI causes nuclear accumulation of the mechanoactivating protein YAP, together with multiple EMT-promoting proteins (i.e., SNAIL, SOX9, HIF2A), we wondered whether mechanoactive cellular characteristics would develop (***Figure 3A***). To answer this question, we first validated that nuclear localization of YAP occurs during NEI across substrate stiffnesses (***Figure 3D, F***). Confirmation of elevated nuclear YAP signified that mechanoactivation might be present independent of substrate triggers. Interestingly, we also found that the shift toward nuclear localization of YAP occurred in both leader and follower cells. Thus, while NEI preserved the intercellular adhesion gradients customary to leader and follower populations, the associated mechanoactivation gradients dissolved.

We considered how this might change cell morphology, phosphorylated myosin light chain (pMLC) expression, and filamentous actin (F-actin). During migration, we found, with high frequency, that NEI caused leader cells to protrude as thin extensions from the monolayer baseline (***Figure 3D***). Comparatively, control leaders maintained more conventional fan-like shapes, even at high stiffnesses. Contributing to these morphological differences, the actin coherency of protruding cells was higher following NEI (***Figure 3H***); meanwhile, pMLC expression was elevated universally in leaders and followers (***Figure 3G***). Curiously, these results suggest that epithelial characteristics (e.g., reinforced

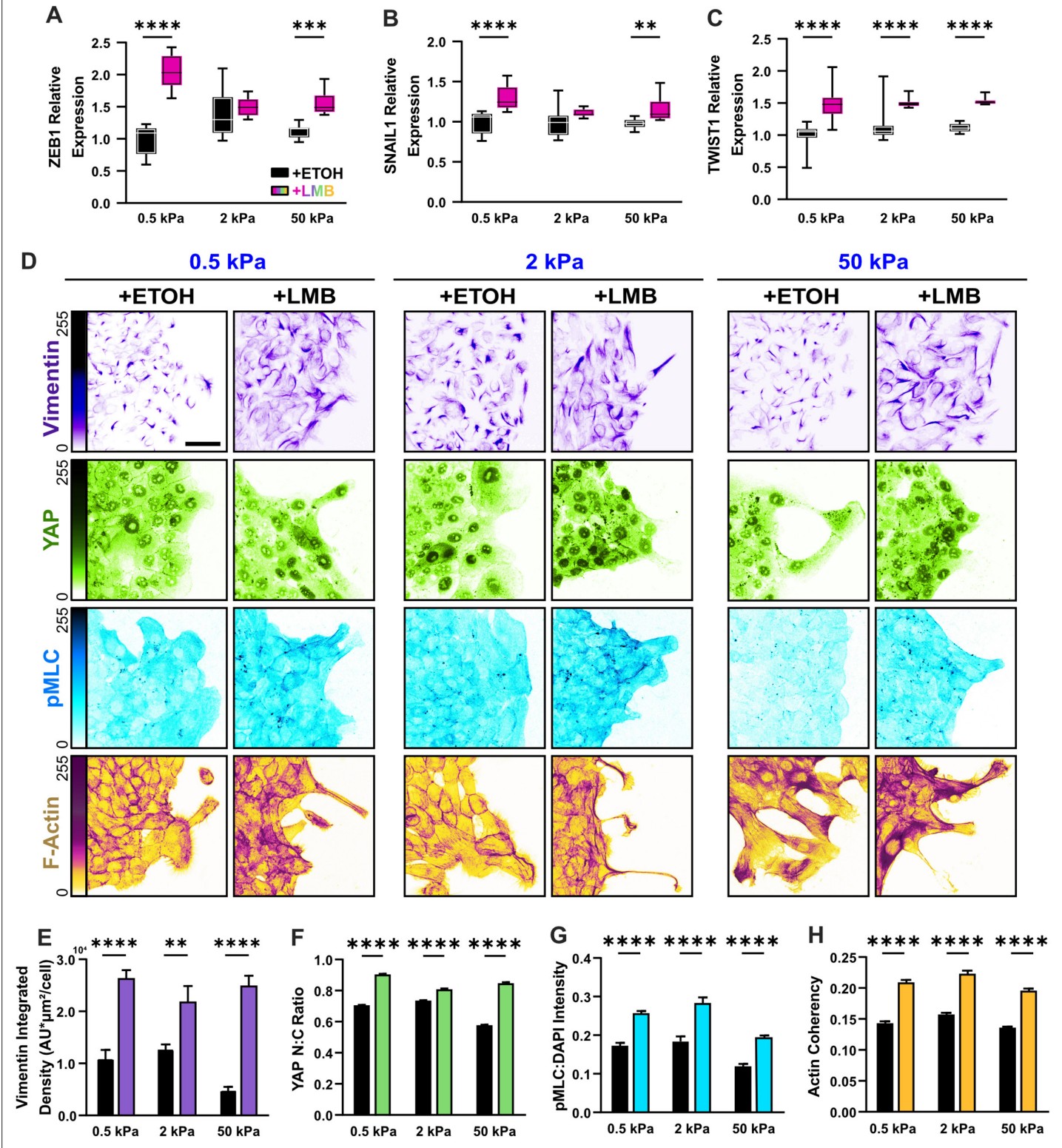

**Figure 3.** Mechanoactive and mesenchymal features develop in MCF10A epithelia during nuclear export inhibition (NEI). (A–C) Relative gene expression for mesenchymal markers ZEB1, SNAIL1, and TWIST1 on 0.5, 2, and 50 kPa polyacrylamide gels. (D) Representative images for mesenchymal characteristics of WT MCF10A across stiffness. Monolayers were treated with ethanol (ETOH) as vehicle or leptomycin B (LMB) for NEI. Images depict vimentin expression, nucleocytoplasmic localization of YAP, phosphorylated myosin light chain (pMLC) expression, and F-actin. (E) Vimentin integrated density, expressed per cell (n = 10), (F) nucleocytoplasmic (N:C) ratio for YAP (n > 55 leaders and 220 followers), (G) pMLC intensity (n = 8), and (H) actin

*Figure 3 continued on next page*

*Figure 3 continued*

coherency (*n* > 1000) for all stiffnesses. Data were analyzed using a two-way analysis of variance (ANOVA) with Tukey post hoc analyses to evaluate NEI and stiffness effects. Significance levels: ** < 0.0021, *** < .0002, **** < 0.0001. Bars represent mean ± standard error of the mean (SEM). Scale bar: 50 μm.

The online version of this article includes the following figure supplement(s) for figure 3:

**Figure supplement 1.** Nuclear export inhibition (NEI) increases cell-generated tractions.

**Figure supplement 2.** The alternate nuclear export inhibition (NEI)-promoting drug, Selinexor, reproduces the concurrent phenotype in WT MCF10A cells.

**Figure supplement 3.** Distinct cell types undergo diverse E-M responses to nuclear export inhibition (NEI).

intercellular adhesions) and mechanoactive features (e.g., higher cytoskeletal forces, front–back polarization) evolve together during NEI.

To assess whether these mechanoactive features extended to traditional mesenchymal markers, we surveyed expression of the EMT TFs, Zinc Finger E-Box Binding Homeobox 1 (ZEB1), Snail Family Transcriptional Repressor 1 (SNAIL1), and Twist Family BHLH Transcription Factor 1 (TWIST1). We also assessed changes in vimentin, which contributes to EMT by promoting cytoskeletal reorganization and focal adhesion formation (*Liu et al., 2015*). Finally, we measured cell traction, where higher tractions indicate mesenchymal-like function (*Mekhdjian et al., 2017*). We found that NEI increased all these mesenchymal markers. ZEB1, SNAIL1, and TWIST1 were upregulated on 0.5 and 50 kPa polyacrylamide gels, while TWIST1 expression increased significantly on 2 kPa gels (*Figure 3A–C*). Therefore, while prior studies found that ZEB1 and SNAIL1 negatively regulate PTEN expression (*Fedorova et al., 2022*), interestingly, these findings indicate that interference with NFκB, or other signaling changes due to NEI, enables simultaneous expression of all these factors. Vimentin expression increased on all substrate stiffnesses (*Figure 3D, E*); and while control monolayers exhibited the characteristic pattern of high traction at the leading edge and lower traction closer to the monolayer core, LMB empowered both leader and follower cells to generate high tractions (*Figure 3—figure supplement 1*). This result indicates that the changes in gene and protein expression after NEI support mesenchymal-like changes in cell function. All taken together, these results demonstrate that NEI elevates mesenchymal characteristics in concert with epithelial ones, indicating the presence of a concurrent epithelial–mesenchymal state.

To determine whether this concurrent E-M state was exclusive to our method of induced NEI (i.e., treatment with LMB), we examined epithelial and mesenchymal characteristics in WT MCF10A cells treated with Selinexor. Treatment with this NEI-promoting drug similarly fostered elevated epithelial and mesenchymal cellular traits (*Figure 3—figure supplement 2*). This suggests that NEI may generally be capable of instituting concurrent E-M phenotypes. Meanwhile, to assess whether NEI could induce concurrent E-M states in other cell types, we examined epithelial and mesenchymal characteristics in MDCK I and MDCK II cells treated with LMB. MDCK I cells exhibited a more typical EMT response, with mesenchymal-like cells clearly separating from the collective group (*Figure 3—figure supplement 3*). On the other hand, MDCK II findings were consistent with a mesenchymally shifted concurrent phenotype, where cells exhibited prominent mesenchymal characteristics while retaining more subtle epithelial ones. Overall, these results from MCF10A, MDCK I, and MDCK II cells indicate that NEI can foster a diverse range of cellular responses. While prior studies in cancer cells found only mesenchymal-epithelial transition (MET) after NEI, interestingly, our findings indicate the potential for a range of concurrent E-M phenotypes, as well as EMT. Ultimately, the capacity for NEI to drive such diverse cellular responses indicates that the phenotypic outcome may depend on the initial proteomic profile of the cells being studied.

## NEI invariably leads to migratory disorder, but changes in migration speed depend on substrate stiffness

Because epithelial cells balance intercellular adhesion and mechanoactivation gradients to facilitate collective movement, we hypothesized that the disturbances observed in MCF10A cells could interfere with typical patterns of migration. To assess these potential migration changes, we collected time-lapse images of migrating MCF10A epithelia for 24 hr, then used particle image velocimetry to determine how NEI affects conventional patterns of migration.

Results reveal stiffness-dependent changes in migration speed, velocity, and order. Previous studies have exclusively shown that NEI inhibits migration; yet, strikingly, on 0.5 kPa, NEI elevated collective migration velocity (*Figure 4A, D*). Higher cellular speeds within the monolayer facilitated this elevation of net migration (*Figure 4G*). Meanwhile, the rise of cell backtracking and overall disorder contributed to the decrease of net velocity toward vehicle levels, beginning around 3 hr (*Figure 4H, M*).

In contrast, on 2 and 50 kPa, NEI decreased net migration velocity at all time points (*Figure 4E, F*). For 2 kPa, this decrease was mediated in initial stages by reduced cellular speeds (*Figure 4I*). Meanwhile, the progressive loss of order and associated increase in cell backtracking exacerbated the overall velocity loss with time (*Figure 4J, N*). For 50 kPa, early time points (0–6 hr) showed only slight decreases in net migration velocity (*Figure 4F*). Higher cell backtracking (*Figure 4L*) accounted primarily for the net migration slowing, since cell speeds were only marginally lower compared to vehicle control (*Figure 4K*), and order was temporarily preserved (*Figure 4O*). Across time points, cell speeds on 50 kPa were comparable to vehicle. Therefore, 50 kPa results resemble 0.5 kPa, where, although speeds were higher, they remained similar to vehicle control. Overall, despite only marginal losses in cell speed on 50 kPa, backtracking together with rising disorder facilitates the observed decrease in collective velocity. Taken as a whole, these findings depict changes to the conventional patterns of collective migration during NEI, where disorder and cell backtracking escalate, independent of substrate stiffness. These results further uncover stiffness-dependent NEI outcomes, where, counter to reports of migration inhibition exclusively, NEI promotes migration on soft substrates.

## Disruption of Golgi polarization in leaders and global elevation of gm130 expression underlie NEI-induced disorder

The Golgi apparatus contributes to front–rear polarization by orienting in front of the nucleus in the direction of migration (*Ravichandran et al., 2020*). Moreover, the associated transport of proteins and nucleation of microtubules toward the front of migrating cells supports directed motion (*Yadav et al., 2009*). To determine whether changes in the Golgi influenced rising disorder during NEI, we analyzed Golgi Matrix Protein gm130, a Golgi marker that assists microtubule nucleation and the development of mesenchymal cellular phenotypes (*Rivero et al., 2009*; *Baschieri et al., 2014*). First, we assessed whether orientation of the Golgi changed in leader and follower cells. To do this, we defined a polarized Golgi as one where the Golgi centroid resided within 60° of the direction of migration, when measured from the associated nuclear centroid (*Figure 5A*; *Mason et al., 2019*). For epithelia treated with vehicle, we found that stiffer substrates promoted leader cell Golgi polarization. Almost twice as many leaders displayed a polarized Golgi apparatus at 50 kPa compared to 0.5 kPa (*Figure 5B, C*). This is consistent with stiffness-induced cell polarization, which has been reported previously (*Alert and Trepat, 2020*). Meanwhile, followers treated with vehicle exhibited a biphasic polarization curve, where 2 kPa had the highest percentage of polarized cells (*Figure 5D*).

NEI's most prominent effect on Golgi orientation was in leader cells. While leader cells customarily provide directional cues for followers, after NEI, the stiffness-dependent polarization of leaders disappeared. Moreover, leader polarization percentages dropped below those reported for control cells. This finding was consistent with the cyclic protrusion and retraction of leader cells observed during migration with NEI and presumably contributed to leading-edge instability. It also reflects previous reports where the rearward position of the Golgi emerged after transition to a mesenchymal cellular state (*Natividad et al., 2018*). Meanwhile, the effect of NEI on follower polarization was more subtle and changed with substrate stiffness.

On 0.5 and 50 kPa, NEI increased follower polarization. This increased Golgi polarization appropriately corresponded to the higher junctional stability, velocities, speeds, order, and backtracking observed during migration. Congruently, the lower polarization on 2 kPa corresponded to reduced migration parameters (*Figure 4*). However, while NEI's impact on Golgi orientation changed with substrate stiffness, its effect on gm130 expression itself was consistent – displaying increased levels across stiffnesses (*Figure 5E*). Together, these results suggest that, in this system, gm130 expression may support the cytoskeletal rearrangement required for the front–back mesenchymal-like polarization observed in leaders, and Golgi orientation itself may contribute to the loss of order observed during NEI.

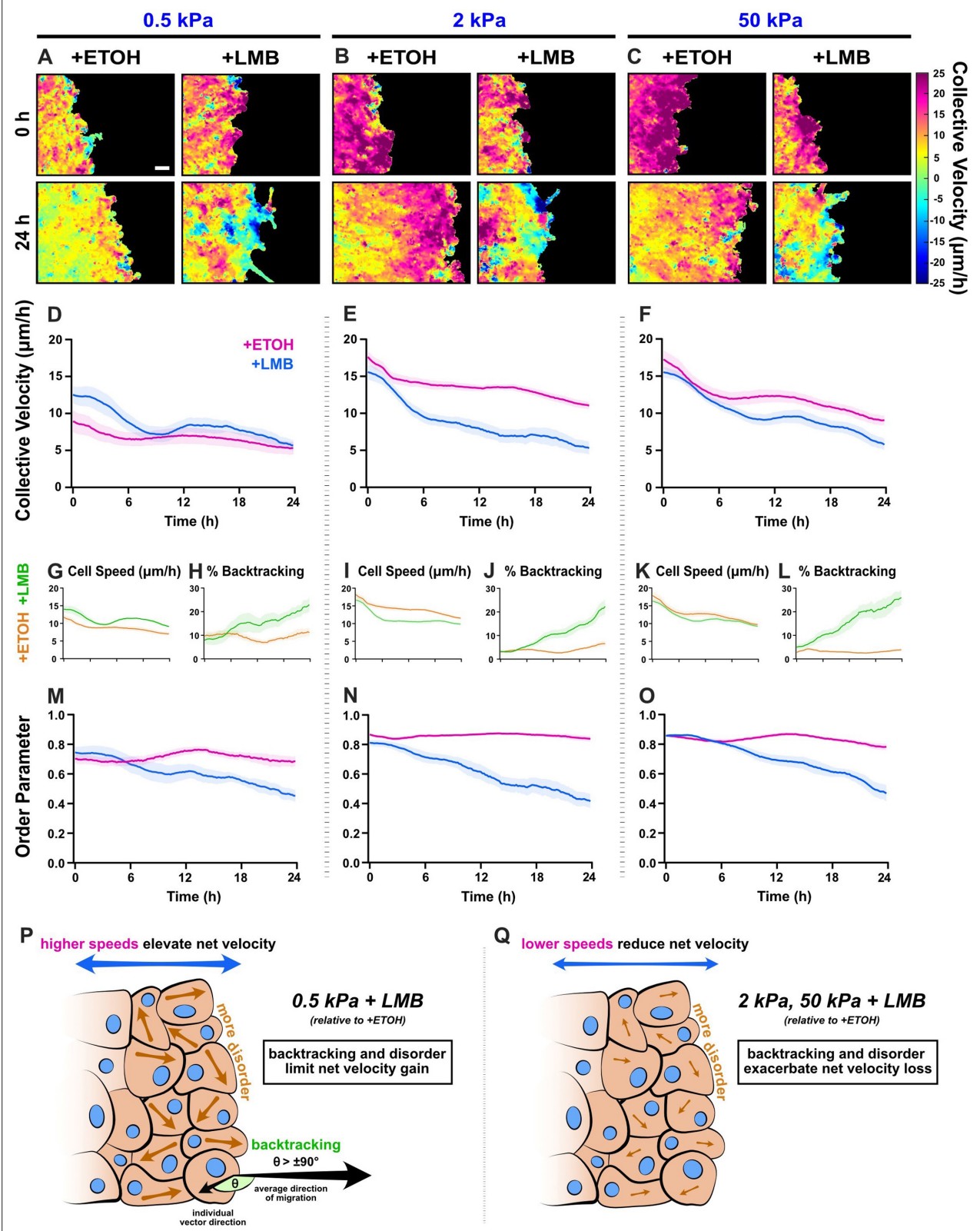

**Figure 4.** Collective migration changes from nuclear export inhibition (NEI) are stiffness dependent. Velocity heat maps generated from particle image velocimetry analysis of 24 hr migration for (**A**) 0.5 kPa, (**B**) 2 kPa, and (**C**) 50 kPa ($n \geq 6$). (**D–F**) Respective quantifications of net velocity over time. Average speeds for (**G**) 0.5 kPa, (**I**) 2 kPa, and (**K**) 50 kPa, along with the % of backtracking vectors (**H, J, L**), respectively. Order parameter for (**M**) 0.5 kPa, (**N**) 2 kPa, and (**O**) 50 kPa. Schematics describing how NEI changes migration characteristics for (**P**) 0.5 kPa and (**Q**) 2 and 50 kPa. Lines represent mean ± standard error of the mean (SEM). Scale bar: 100 µm.

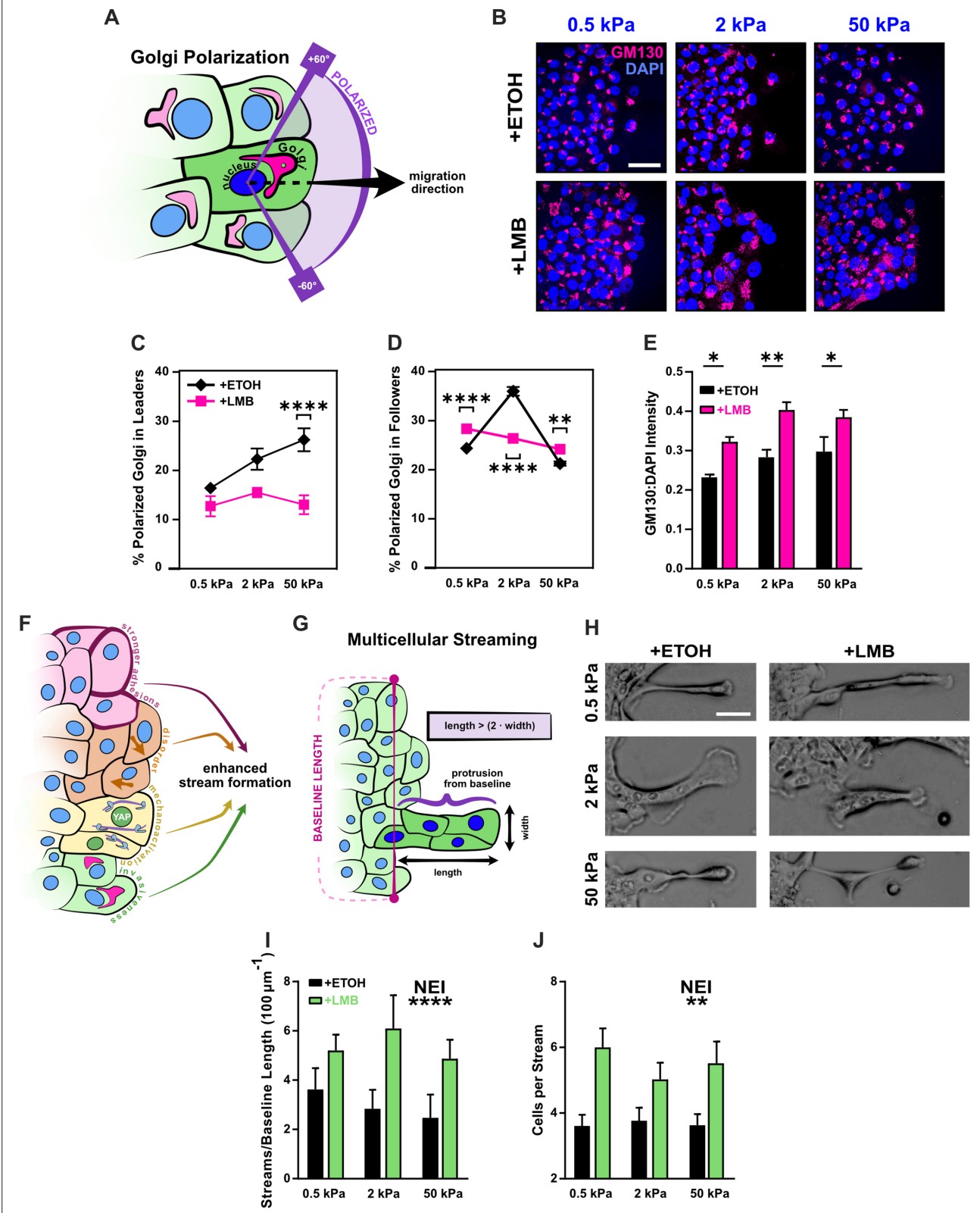

**Figure 5.** Disrupted Golgi polarization and GM130 expression underlie disordered migration while the combined epithelial–mesenchymal features support multicellular streaming. (**A**) Schematic definition for Golgi polarization, where a polarized Golgi is one whose centroid lies within 60° of the line drawn from the nuclear centroid in the average direction of migration. (**B**) Representative images for gm130 and DAPI across substrate stiffnesses and cell treatments. Quantification of Golgi polarization in (**C**) leaders ($n > 50$) and (**D**) followers ($n > 250$). Shapes represent mean and error bars represent

*Figure 5 continued on next page*

*Figure 5 continued*

standard error of the mean (SEM). (**E**) Quantification of gm130 intensity across substrate stiffnesses (*n* = 8). (**F**) Schematic depicting nuclear export inhibition (NEI)-induced changes in cell phenotype and migration, which imply the potential for enhanced stream formation. (**G**) Schematic definition for multicellular streams. (**H**) Example images for multicellular streams seen across stiffnesses and treatment conditions. Accompanying quantification for the number of (**I**) streams per baseline length and (**J**) cells per stream. Bars represent mean ± SEM. Data were analyzed using two-way analyses of variance (ANOVAs) with Tukey post hoc analyses to evaluate NEI and stiffness effects. *'s denote the significance level for NEI effects. Significance levels: * < .0332, ** < 0.0021, *** < .0002, **** < 0.0001. Scale bar: 50 μm.

## Leading-edge instability and combined epithelial–mesenchymal characteristics promote multicellular stream formation during NEI

Previous studies established that leading-edge instability, higher intercellular adhesion, localized mechanoactivation, and coordinated cellular velocities together support the formation of multicellular streams (*Vishwakarma et al., 2018*; *Sarker et al., 2019*). Here, we have shown that NEI destabilizes the leading edge, demonstrated via decreased leader polarization; reinforced intercellular adhesion, measured via increased junctional p120 and ZO-1; caused mechanoactivation, evidenced via higher YAP N:C ratio, pMLC expression, and F-actin coherency; and contributed invasive characteristics, shown through elevated expression of gm130 and vimentin. Therefore, despite the reduced overall coordination of velocity during NEI, reflected in low order parameter, we hypothesized that, at some level, NEI might encourage multicellular streaming (*Figure 5F*). To investigate this hypothesis, as previously, we defined a cell stream as a protrusion of two or more cells in succession from the mono-layer baseline, such that protrusion length was more than twice its width (*Figure 5G*; *Sarker et al., 2019*). After examining migration time-lapses, we found that transient cell streams formed, even during migration with vehicle (*Figure 5H*). However, these streams formed at relatively low frequency (*Figure 5I*) and with few cells (*Figure 5J*). The presence of streams in this control condition suggests that fluctuations in intercellular adhesion and mechanoactivation at the leading edge were sufficient to form small, transient streams. By contrast, streams during NEI were more exaggerated (*Figure 4A–C* and *Figure 5H*), evolving at higher frequency and with more cells in each stream (*Figure 5I, J*). Higher stream formation during NEI thus indicates that adhesion strength and mechanoactivation may stimulate brief periods of cell streaming, even in regions of high disorder. Therefore, while previous studies show that structural changes in the underlying matrix (e.g., collagen fiber orientation, topography) trigger cell streaming via adhesion-based signaling and consequent mechanoactive protein expression (*Vishwakarma et al., 2018*; *Sarker et al., 2019*), these results highlight that streaming can also arise from direct modifications to nuclear export.

## YAP silencing shifts NEI-affected cells toward an epithelial state by strengthening intercellular adhesion and attenuating mechanoactivation

NEI causes nuclear localization of YAP, which generally promotes mechanoactive phenotypes, such as that previously shown (*Figure 3*). Given its role as an EMT initiator (*Park et al., 2019*), we hypothesized that excluding YAP from the affected cargos could reduce mesenchymal characteristics and shift cells toward a more conventional epithelial state. We examined the effect of YAP silencing across 0.5, 2, and 50 kPa stiffnesses (*Figure 6—figure supplements 1–3*). However, to discuss the core differences observed during NEI without YAP, we focus on results from the softest substrate, 0.5 kPa (*Figure 6*).

In contrast to WT cells, where IκBα localized cleanly within the cytoplasm or nucleus, shYAP cells exhibited punctate staining at intercellular adhesions in control conditions (*Figure 6—figure supplement 1E*) and displayed strong cytoplasmic signals during NEI (*Figure 6A*). These differences are consistent with IκBα degradation, a known by-product of YAP silencing (*Wang et al., 2020*). However, YAP silencing significantly increased p120 expression, indicating stronger intercellular adhesion than WT for all conditions (*Figure 6A–C*). This finding suggests that removing YAP from the affected cargos enhances epithelial aspects of NEI's concurrent E-M phenotype.

To determine how YAP silencing affected mesenchymal characteristics, we next examined differences in pMLC, F-actin, gm130, and cell morphology during NEI. Interestingly, pMLC levels after NEI were unchanged with YAP silencing (*Figure 6D, E*). However, YAP-silenced cells exhibited reduced actin coherency and gm130 expression relative to WT cells (*Figure 6F, G*). Meanwhile, NEI-related

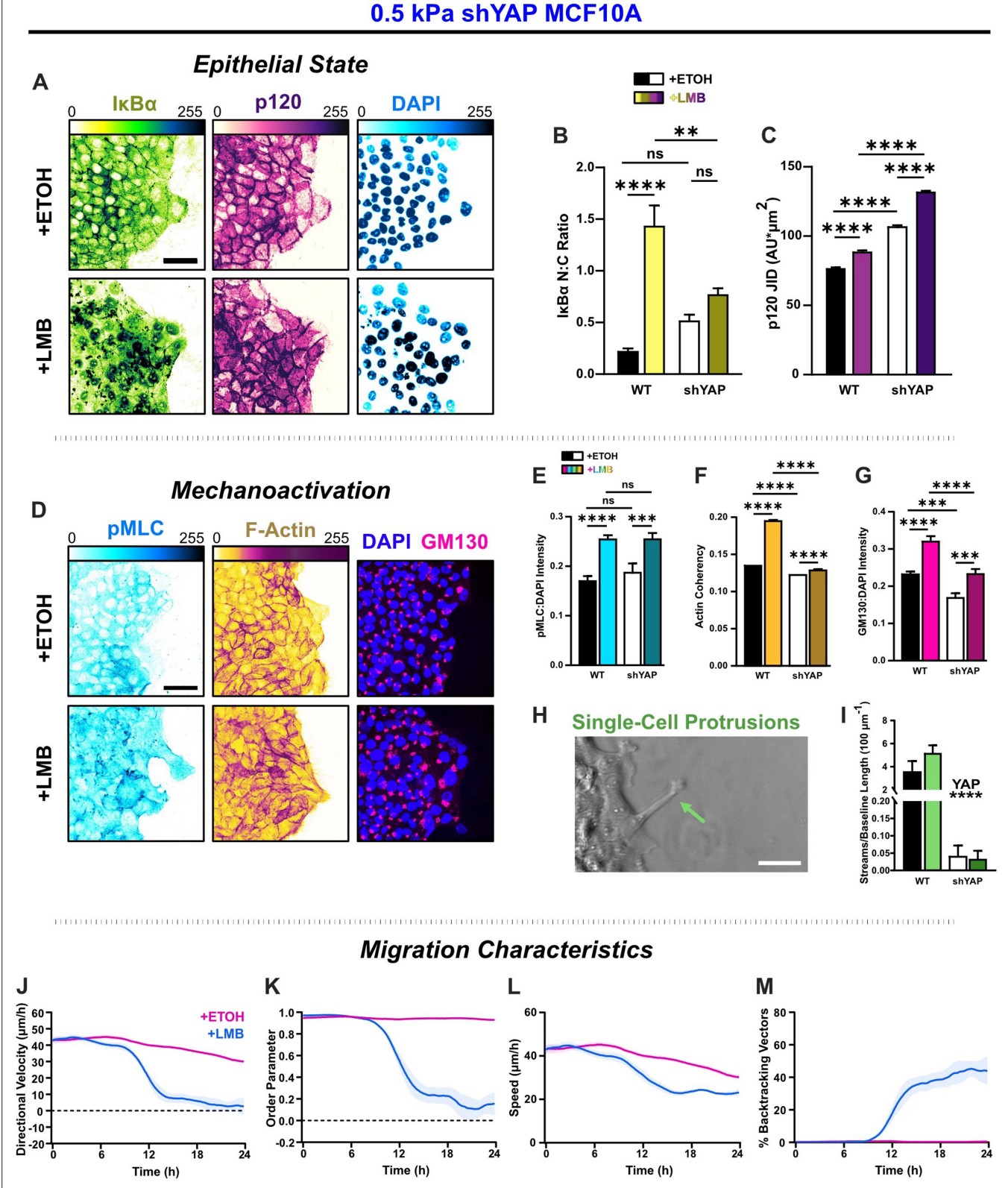

**Figure 6.** YAP silencing enhances epithelial characteristics and attenuates mechanoactivation. (**A**) Epithelial characteristics for shYAP MCF10A. Images depict nucleocytoplasmic localization of IκBα (left), p120 expression (middle), and DAPI nuclear signal (right). (**B**) Nucleocytoplasmic (N:C) ratio for IκBα ($n$ = 8). (**C**) Changes in p120 junction integrated density (JID) ($n$ = 8). (**D**) Mechanoactive characteristics for shYAP MCF10A. Images depict phosphorylated myosin light chain (pMLC) (left), F-actin (middle), and gm130 expression (right). Quantification comparing WT and shYAP (**E**) pMLC

*Figure 6 continued on next page*

Figure 6 continued

expression ($n$ = 8), (**F**) actin coherency ($n$ > 1000), and (**G**) gm130 expression ($n$ = 8). (**H**) Representative image showing single-cell protrusions at the leading edge. (**I**) Quantification of multicellular streams per baseline length for WT and shYAP cells ($n \geq 6$). Bars represent mean ± standard error of the mean (SEM). All data were analyzed using two-way analyses of variance (ANOVAs) with Tukey post hoc analyses to evaluate nuclear export inhibition (NEI) and YAP effects. For streams, *'s denote the significance level for YAP effects. Significance levels: ** < 0.0021, *** < .0002, **** < 0.0001. Time plots of migration characteristics ($n \geq 6$): (**J**) net velocity, (**K**) order parameter, (**L**) speed, and (**M**) percentage of backtracking vectors for shYAP monolayers on 2 kPa polyacrylamide gel. Lines represent mean ± SEM. Scale bars: 50 µm.

The online version of this article includes the following figure supplement(s) for figure 6:

**Figure supplement 1.** Epithelial characteristics for shYAP cells for all stiffnesses.

**Figure supplement 2.** Mechanoactive characteristics for shYAP cells for all stiffnesses.

**Figure supplement 3.** shYAP migration characteristics across stiffnesses.

changes in gm130 expression were insignificant. Consistent with this reduction in the cytoskeletal features required for mesenchymal presentation, YAP-silenced leaders formed much shorter protrusions at the leading edge (*Figure 6H*), and multicellular streams were also inhibited (*Figure 6I*). Together, these results demonstrate that isolating YAP from affected cargos attenuates mesenchymal aspects of the combined E-M phenotype induced during NEI.

This shift in E-M state led to significant changes in collective migration. For NEI on 0.5 kPa, YAP-silenced cells exhibited sudden drops in net migration velocity beginning around 12 hr. Migration on 2 and 50 kPa exhibited similar drops, which preceded the complete arrest of net migration for all stiffnesses (*Figure 6—figure supplement 3A–C*). Moreover, at later time points, cells transiently reversed their direction of migration (*Figure 6J*, *Figure 6—figure supplement 3A–C*). Results suggest that changes in migration order prompted those changes in velocity. At 12 hr, the sudden drop in net velocity coincided with a sudden drop in migration order, and at later time points, migration velocity, and order parameter jointly oscillated about zero (*Figure 6J, K*, *Figure 6—figure supplement 3A–C, J–L*). Interestingly, and contrary to WT, YAP-silenced cells exhibited non-monotonic increases in cell backtracking (*Figure 6—figure supplement 3E, G, I*). Inflection points for backtracking coincided with inflection points for net velocity, order parameter, and speed. These points were present across substrate stiffnesses and reflected transient oscillations in order within the monolayer (*Figure 6—figure supplement 3*). In general, the observed worsening of net velocity and order losses during NEI is consistent with previous findings of impersistent cell motility following YAP depletion (*Mason et al., 2019*). Therefore, while YAP contributes to mechanoactive features during NEI, it also provides crucial directional cues to assist collective migration. In other words, YAP may enhance cohesion to temper the loss of order during NEI.

Notably, NEI initially increased shYAP cell speeds on 0.5 and 2 kPa, which could indicate residual YAP-independent mechanoactivation. However, after 6 hr, speeds were lower than vehicle control for all substrate stiffnesses (*Figure 6—figure supplement 3D, F, H*). This time dependency could indicate that early NEI changes are dominated by mechanoactive behavior, while cell–cell adhesion changes occur later. Overall, these results demonstrate that YAP silencing shifted NEI's concurrent E-M state by enhancing epithelial aspects and attenuating mesenchymal ones.

## α-Catenin knockdown shifts NEI-affected cells toward a mesenchymal state but prevents the transfer of mechanoactive features to collective behaviors

Propagation of forces through adherens junctions is an element fundamental to collective migration (*Ladoux and Mège, 2017*). One of the central mediators in this cooperative force propagation serving to mechanically couple cells is the protein α-catenin, which recruits F-actin to E-cadherin-based cell junctions. Therefore, to characterize the role of intercellular force propagation in the observed NEI outcomes, we used α-catenin depleted MCF10A cells, as described previously (*Loza et al., 2016*). We examined the effect of α-catenin knockdown across 0.5, 2, and 50 kPa stiffnesses (*Figure 7—figure supplements 1–3*). However, to discuss the key outcomes from α-catenin depletion, we again focus on results from the softest substrate, 0.5 kPa (*Figure 7*).

By itself, α-catenin knockdown (α-cat KD) destabilized intercellular adhesions and yielded a mesenchymal phenotype (*Figure 7A, D*). Cells at the leading edge best epitomized mesenchymal

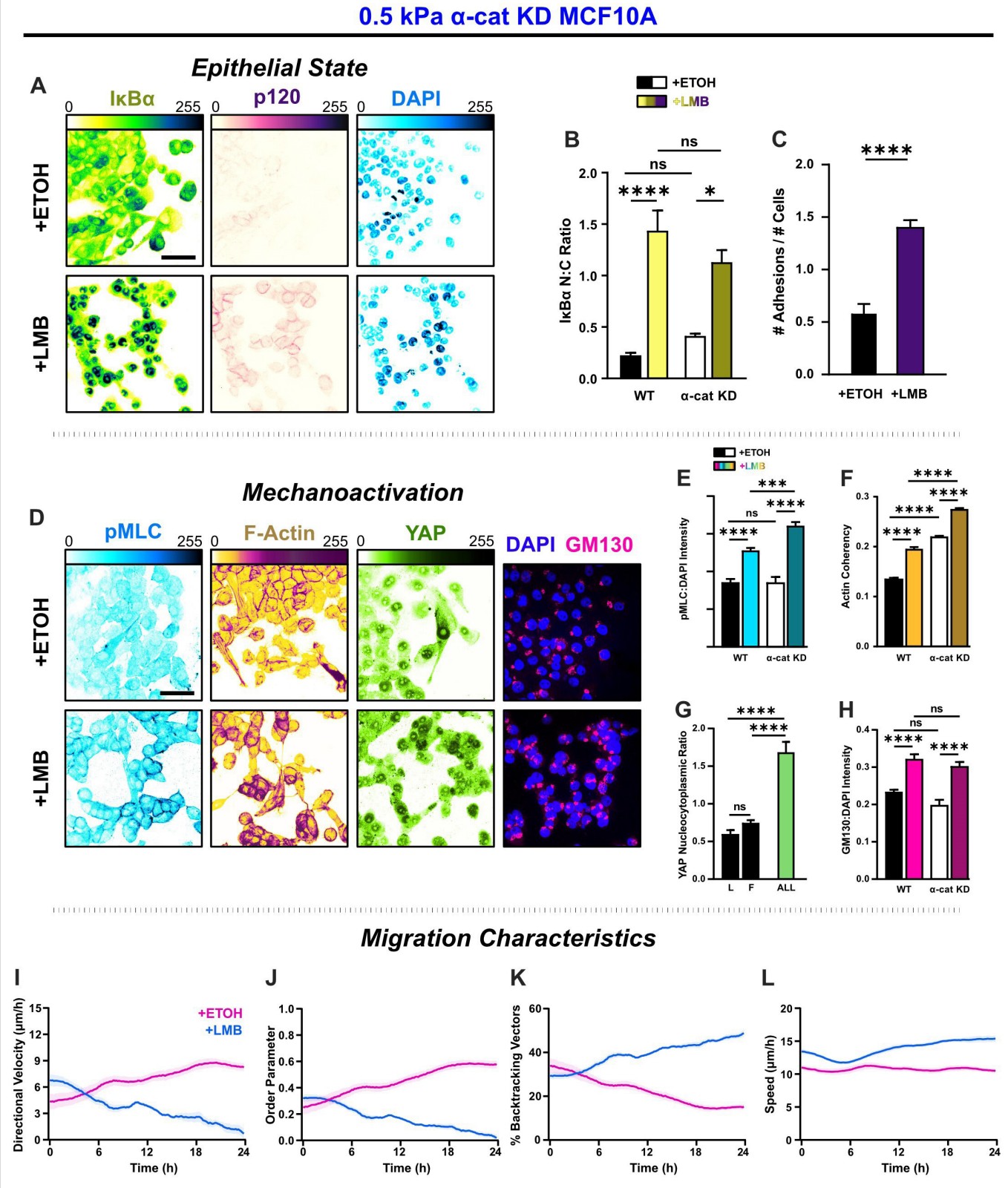

**Figure 7.** α-Catenin knockdown prevents the transfer of mechanoactive features to collective cell behavior. (**A**) Epithelial characteristics for α-cat KD MCF10A. Images depict nucleocytoplasmic localization of IκBα (left), p120 expression (middle), and DAPI nuclear signal (right). (**B**) Nucleocytoplasmic (N:C) ratio for IκBα (n = 8). (**C**) Changes in the number of discernible p120-marked junctions (n = 8). (**D**) Mechanoactive characteristics for α-cat KD MCF10A. Images depict phosphorylated myosin light chain (pMLC) (left), F-actin (middle), and gm130 expression (right). Quantification comparing WT

*Figure 7 continued on next page*

*Figure 7 continued*

and α-cat KD (**E**) pMLC expression (*n* = 8), (**F**) actin coherency (*n* > 1000), (**G**) nucleocytoplasmic (N:C) YAP ratio, and (**H**) gm130 expression (*n* = 8). Bars represent mean ± standard error of the mean (SEM). All data were analyzed using two-way analyses of variance (ANOVAs) with Tukey post hoc analyses to evaluate nuclear export inhibition (NEI) and α-catenin effects.Significance levels: * < .0332, *** < .0002, **** < 0.0001. Time plots of migration characteristics (*n* ≥ 6): (**I**) net velocity, (**J**) order parameter, (**K**) percentage of backtracking vectors, and (**L**) speed for α-cat KD monolayers on 2 kPa polyacrylamide gel. Lines represent mean ± SEM. Scale bars: 50 μm.

The online version of this article includes the following figure supplement(s) for figure 7:

**Figure supplement 1.** Epithelial characteristics for α-cat KD cells for all stiffnesses.

**Figure supplement 2.** Mechanoactive characteristics for α-cat KD cells for all stiffnesses.

**Figure supplement 3.** α-cat KD migration characteristics across stiffnesses.

characteristics, exhibiting more morphological polarization, aided by their separation from the cells behind. Cells residing closer to the monolayer center remained bunched due to contact inhibition. However, cell–cell adhesion there was still low. NEI generally caused re-epithelialization of α-cat KD cells, during which cells lost front–back polarity, formed discrete colonies, and dissolved leader–follower relationships (*Figure 7A, D*). However, while NEI caused a nearly three times increase in intercellular adhesions (*Figure 7C*) and promoted nuclear accumulation of IκBα (*Figure 7B*), p120 expression was markedly lower than WT for all conditions. These results suggest that interrupting force propagation through intercellular adhesions decreases the epithelial features of NEI's concurrent E-M phenotype.

Without this cooperation between intercellular adhesions and intracellular forces intrinsic to migrating epithelia, cells treated with vehicle migrated as a mesenchymal collective (*Theveneau and Mayor, 2013*). α-cat KD cells migrated with lower speed and order, exhibiting more backtracking and thus lower net velocity relative to WT (*Figure 7I, J*). However, likely arising from contact inhibition, these cells developed order and net velocity over time. Meanwhile, in removing the cooperative force balance, data suggest we also severed the link required to sustain NEI's concurrent E-M state. Without force propagation between cells, the effect of NEI on migration was time dependent across stiffnesses. Initially, NEI elevated net velocity. From 0–4 hr on 0.5 and 50 kPa, net velocities were higher than vehicle control. On 2 kPa, this initial phase extended to 8 hr. After that phase, velocities decreased and approached zero by 24 hr (*Figure 7—figure supplement 3A–C*). Differences in order underlay the differences in velocity. During the initial time points, cells migrated with slightly higher order and speed (*Figure 7L*) and thus achieved higher collective velocity. However, while higher speeds endured for the time-lapse duration, there was progressive disarray and loss of net velocity. Cell backtracking mirrored these changes (*Figure 7K*). Overall, findings suggest that α-catenin knockdown exacerbates the loss of order during NEI to facilitate the arrest of net migration. In other words, intercellular adhesion reinforcement and YAP may jointly enhance cohesion to slow the loss of order during NEI. Outcomes from removing α-catenin further imply an unusual partnership paramount to the concurrent E-M phenotype, where stronger intercellular adhesion facilitates the propagation of higher forces through the collective to sustain the simultaneous E-M state.

## Discussion

Classically, epithelial (E) and mesenchymal (M) phenotypes exist along a single continuum, where cells traverse from E to M via the gradual loss of epithelial characteristics and the accompanying gain of mesenchymal ones. At the same time, it is understood that EMT is highly complex. The broad array of TFs and RNA regulators involved in EMT give rise to vast diversity in EMT programs. Given this diversity, it is accepted that some biological markers may appear in conflict with one another for simple EMT categorization. Acknowledging this, groups have emphasized that single biological markers are not sufficient to categorize cellular states and that functional cellular changes should take precedence for appropriate categorization of E-M phenotypes (*Yang et al., 2020*). However, together, these statements pose a compelling question about the interaction between EMT-promoting and -inhibiting TFs during the development of cell phenotypes: do these proteins inherently compete, cooperate, or cancel? Here, we use LMB, a CRM1-based inhibitor of nuclear export, to explore this question. Our findings indicate that EMT-promoting and -inhibiting pathways can elevate epithelial

and mesenchymal features simultaneously (indicating some degree of cooperation), but results also suggest that cell-level epithelial and mesenchymal features compete, which gives rise to disordered epithelia.

Across substrate stiffnesses, LMB-treated MCF10A collectives simultaneously elevate expression of epithelial and mesenchymal marker genes. Leader and follower cells exhibit intercellular adhesion strengthening, mediated by p120 and ZO-1. However, in spite of these stronger intercellular adhesions, leaders develop highly polarized morphology, and their protrusions contain more coherent actin filaments. Mechanoactivation markers: YAP, pMLC, vimentin, and gm130 expression globally increase, while follower cells generate stronger tractions. Ultimately, within the collective, cells manifest atypical E-M states, in which elements at opposite ends of the conventional spectrum coexist (*Figure 8C*). Interestingly, these results suggest the classical view that epithelial and mesenchymal phenotypes are purely antagonistic – and accordingly exist along a single continuum (*Figure 8A*.I) – could be incomplete.

Instead, our findings from NEI suggest that opposing transcriptional programs can work simultaneously to increase epithelial and mesenchymal characteristics, which we refer to here as concurrent E-M. Within such simultaneous states, we also demonstrate that epithelial and mesenchymal characteristics can individually vary. We show that excluding YAP from the NEI-affected cargos attenuates some mesenchymal phenotypic components while strengthening epithelial ones (*Figure 8D*). We also show that, in the absence of p120, MDCK II cells significantly increase mesenchymal characteristics in response to NEI, while adhesions are maintained (*Figure 8F*). The existing conceptual E-M model provides for the loss of epithelial traits precisely in step with the gain of mesenchymal ones (i.e., EMT), and for the reverse process (i.e., MET). However, it is unable to explain the concurrent states we observe. Therefore, our findings indicate that an expanded model may be necessary to fully capture the range of biological phenotypes. Here, we propose a model where epithelial and mesenchymal features can traverse separate continua (*Figure 8A*.II). We allocate the terms epithelial and mesenchymal transducers to represent the collection of E-M proteins within the cell, and propose that they strongly drive their respective phenotype, indicated by thick links. Meanwhile the opposite transducer prevents the opposite phenotype only weakly, indicated by the thin links. Ultimately, this change to the model accommodates EMT phenotypes as they stand: epithelial, partial EMT, and mesenchymal (*Figure 8B*.I–III), but allows for varying degrees of concurrent E-M features (*Figure 8B*.IV–VII).

This model expansion may be needed to explain existing migratory phenomena. For example, one problem fundamental to collective migration is how follower cells maintain high adhesion to their neighbors while simultaneously extending cryptic protrusions to assist grouped migration (*Lu and Lu, 2021*). Meanwhile, jointly elevated epithelial and mesenchymal signatures have been observed in chemo-resistant and aggressive cancer cells (*Xuan et al., 2020*). Taking these phenomena as an example, investigating migratory phenotypes within a framework that allows for simultaneous enrichment of epithelial and mesenchymal features, like that we observed here, may provide a more complete understanding of key biological processes and assist the identification of appropriate therapeutic options.

Our results indicate that signaling pathways may cooperatively elevate epithelial and mesenchymal traits, but they also suggest that epithelial and mesenchymal traits are themselves antagonistic. It's been shown that higher cell–cell adhesion predisposes cells to repolarization cues from neighbors and reduces the contribution of polarization stemming from mechanosensitive adhesion-based promotion of locomotion (*Desai et al., 2013*; *Qin et al., 2021*). Typically, leader cells exhibit lower intercellular adhesion, which allows them to polarize front–back and generate most of the traction that propels the collective (*Reffay et al., 2014*; *Trepat and Sahai, 2018*). Meanwhile, follower cells generally retain higher cell–cell adhesion and put forth lower traction. NEI jumbles these leader–follower phenotypes, fostering a collective where all cells have stronger intercellular adhesions, which makes them more subject to repolarization from their neighbors; and all cells generate high traction, which also allows them to more individually drive migration and contribute to repolarization of their neighbors. Across substrate stiffnesses, what results is progressively disordered cell migration during NEI. From a broader standpoint, these results emphasize that epithelial and mesenchymal characteristics can exist in high degrees concurrently but not necessarily cooperatively.

Results also suggest that the simultaneous presence of these competing characteristics may be facilitated by adherens junctions. Our experiments demonstrated that knockdown of α-catenin

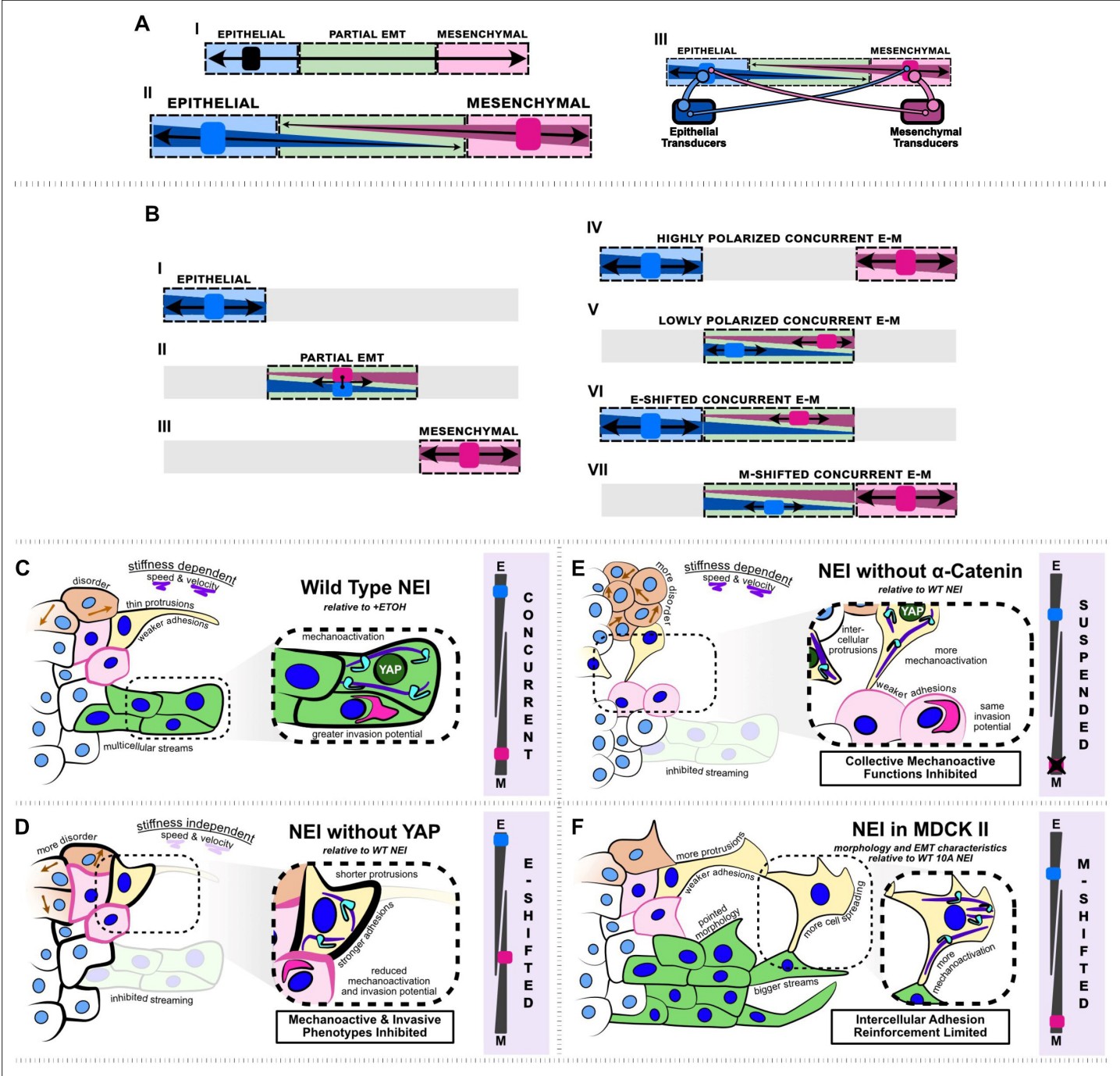

**Figure 8.** Summary schematic depicting nuclear export inhibition (NEI) outcomes for WT MCF10A, shYAP MCF10A, α-catenin knockdown MCF10A, and MDCK II cells. (A.I) Schematic depicting the current understanding of the epithelial–mesenchymal (E-M) continuum, which requires that cells must lose epithelial traits as they acquire mesenchymal ones. (A.II) Simplified schematic depicting our proposed conceptual model where epithelial and mesenchymal traits exist on separate continua, allowing for concurrent E-M states (bottom). (A.III) Schematic depicting the implied conceptual mechanism, whereby individual epithelial and mesenchymal markers may pull the phenotype in separate directions. The individual number and strength of epithelial and mesenchymal transducers determines the ultimate cellular phenotype. (B.I–III) Established epithelial and mesenchymal states shown on the proposed continua. (B.IV–VII) Epithelial and mesenchymal states exhibiting concurrent behavior, shown on the proposed continua. (**C**) Relative to vehicle, WT MCF10A exhibit stronger intercellular adhesions, together with higher mechanoactivation and invasion potential. Cells exhibit thin protrusive morphology as single cells and form more multicellular streams at the leading edge. We propose this is a highly polarized concurrent epithelial–mesenchymal (E-M) state. NEI disrupts collective migration, causing stiffness-dependent changes in speed and velocity, and overall migratory disorder. (**D**) Relative to WT NEI, NEI for shYAP yields stronger intercellular adhesions while lowering mechanoactivation and invasion potential. Cells form shorter protrusions and are unable to form multicellular streams. We propose this is an E-shifted concurrent state. (**E**) Relative to WT NEI, NEI

*Figure 8 continued on next page*

*Figure 8 continued*

for α-cat KD MCF10A yields weaker intercellular adhesions and more mechanoactivation. Cells protrude to form more intercellular connections, but collective mechanoactive function is inhibited. Given the strong mechanoactive signatures, but low mechanoactive behavior, we propose that α-cat KD suspends the concurrent E-M state. (**F**) Relative to WT MCF10A, NEI for MDCK II cells increases mesenchymal characteristics further, with cells exhibiting highly spread and protrusive morphology. Cells exhibit fewer epithelial characteristics in the absence of p120-mediated intercellular adhesion reinforcement; however, cells still retain adhesion and form bigger multicellular streams. We propose this is an M-shifted concurrent state.

The online version of this article includes the following figure supplement(s) for figure 8:

**Figure supplement 1.** Outcomes from WT nuclear export inhibition (NEI) are biphasic with substrate stiffness and may derive from shifts in the balance between intercellular adhesion and mechanoactivation.

suspended the concurrent phenotype (*Figure 8E*). Given α-catenin's role in recruiting F-actin to adherens junctions and instituting tension across cell–cell contacts (*Ladoux and Mège, 2017*), this is perhaps not surprising. It indicates that concurrent E-M states require cooperation between intercellular adhesion and intracellular forces. On the other hand, MDCK cells without junctional p120 exhibited more mesenchymal-like states. Together, these results suggest that adherens junctions contain the components required to resist the concurrent phenotype's collapse to both an epithelial state (i.e., α-catenin) and to classic EMT (i.e., E-cadherin). In conjunction, these results highlight the diversity of cellular outcomes, spanning broadened E-M spectra, following NEI.

Clinically, SINE-based NEI has potential to treat cancer – largely for its ability to re-localize escaped tumor suppressors to the nucleus (*Zhong et al., 2014*; *Kashyap et al., 2016a*; *Kashyap et al., 2016b*; *Galinski et al., 2021*; *Gravina et al., 2014*; *Gravina et al., 2017*; *Turner et al., 2014*). Multiple studies investigating Selinexor have suggested that NEI causes EMT reversal in cancer cells, with inhibition of NFκB by IκBα playing a significant role (*Kashyap et al., 2016a*; *Kashyap et al., 2016b*; *Galinski et al., 2021*). This finding is particularly interesting in light of our findings of mechanoactivation after Selinexor treatment. It may be that NEI differentially affects cells based on their distinct proteomic profiles, as we observed here in MCF10A and MDCK cells. Or, it is plausible that standard tissue culture plastic masks some of NEI's mechanoactive outcomes because of changes in nuclear pore selectivity (*Elosegui-Artola et al., 2017*). Comprehensively surveying mechanoactive and mesenchymal markers, and measuring cellular outcomes on substrates better approximating the stiffnesses of physiological tissues, would answer this question. Ultimately, whether concurrent E-M outcomes manifest in clinical implementations of NEI, and more pointedly, whether mechanoactivation could antagonize therapeutic effects remains to be seen.

To put these results in context, we acknowledge limitations of our study. Ultimately, our experiments sought to determine how EMT-promoting and -inhibiting proteins interact to enact cell phenotypes. We used NEI to implement the complex situation where competing EMT-related proteins were simultaneously maintained in the nucleus. However, CRM1 has ~241 cargos and therefore affects many proteins beyond EMT scope (*Turner et al., 2014*; *Xu et al., 2012*). To account for this nonspecificity, we frame our conclusions as phenotypic outcomes of NEI. For more finely tuned phenotypic manipulations, the localization of E-M proteins would need to be controlled individually. Such future studies may reveal more detailed genomic and proteomic regulation of concurrent E-M phenotypes.

In sum, here we show that NEI promotes biological (gene, protein expression) and functional (morphological, migratory) changes that expand the traditional epithelial–mesenchymal continuum. We propose these changes constitute a concurrent E-M state in which highly epithelial and highly mesenchymal characteristics coexist. With MCF10A YAP knockdown and MDCK II cells, we demonstrate that the concurrent state has an adjustable range, where cells appear more epithelial or more mesenchymal, but retain prominent characteristics of both phenotypes. Meanwhile, results from MCF10A α-catenin knockdown cells indicate that the simultaneous phenotype hinges on cooperation between intercellular adhesion and intracellular force, with adherens junctions emerging as the operative link. Ultimately, this concurrent manifestation expands the classical interpretation of phenotypes, and indicates that epithelial and mesenchymal traits may exist on separate continua. These findings highlight an increasingly diverse range of epithelial–mesenchymal phenotypes and thus further encourage a more comprehensive assessment of E-M cellular features as a basis for phenotypic categorization. Overall, this expanded understanding of E-M states may help inform future studies involving epithelial mechanobiology and pathology.

## Ideas and speculation: *competing forces*

At a basic level, the cohesion of epithelial collectives derives from a balance between intercellular adhesion strength and cell-generated traction forces (*Trepat and Sahai, 2018*). After NEI, the migratory order of epithelia was generally low; however, it appeared to exhibit a biphasic relationship with substrate stiffness (*Figure 8—figure supplement 1*). We observed that cell–cell adhesion strength was lowest on 2 kPa and higher on both the softer and stiffer substrates (*Figure 8—figure supplement 1A*). Meanwhile, mechanoactivation was high across stiffnesses (*Figure 8—figure supplement 1B–E*). These distinct outcomes suggest that NEI may disrupt the balance between intercellular adhesion and cell traction required for collective cohesion. As an illustration of this idea, higher percentage losses in migration speed, velocity, order, and polarization (*Figure 8—figure supplement 1F–K*) correlated with lower adhesion strength (i.e., 2 kPa). This biphasic trend across migration parameters extended to stream formation. The higher disorder on 2 kPa implied that epithelia could encounter greater leading-edge instability, and 2 kPa indeed coincided with a higher protrusion rate of multicellular streams (*Figure 8—figure supplement 1J*). However, despite this increase in stream formation, lower cell cohesion limited the number of cells that successfully coordinated their velocities to establish streams (*Figure 8—figure supplement 1K*). Altogether, the balance between intercellular adhesion and mechanoactivation was most uneven on 2 kPa, and the disorder was worst there. Therefore, these findings suggest that, while NEI enhances both epithelial and mesenchymal traits, cells may not develop strong enough intercellular adhesions to balance cellular tractions, and this imbalance may ultimately contribute to the observed migratory disorder.

In contrast to the disordered migratory patterns of the concurrent phenotype observed after NEI, under normal conditions, follower cells sustain high cooperation between epithelial and mesenchymal features. In our experiments, epithelial and mesenchymal characteristics extended far from the phenotypic center point. However, typical follower cells maintain lower cell–cell adhesion and traction. Therefore, it is possible that the degree of cooperation during concurrent E-M may ultimately depend on the distance of either E-M component from the phenotypic center – that is, the magnitude difference between intercellular adhesive forces and cell-generated tractions.

# Materials and methods

**Key resources table**

| Reagent type (species) or resource | Designation | Source or reference | Identifiers | Additional information |
|---|---|---|---|---|
| Cell line (*Homo sapiens*) | MCF10A breast, epithelial | ATCC | Cat. #: CRL-10317 | N/A |
| Cell line (*Homo sapiens*) | shYAP MCF10A breast, epithelial | Dr. Gregory Longmore Nasrollahi, S., Walter, C., Loza, A. J., Schimizzi, G. v., Longmore, G. D., & Pathak, A. (2017). Past matrix stiffness primes epithelial cells and regulates their future collective migration through a mechanical memory. *Biomaterials*, *146*, 146–155. https://doi.org/10.1016/j.biomaterials.2017.09.012 | N/A | N/A |
| Cell line (*Homo sapiens*) | α-catenin KD MCF10A breast, epithelial | Dr. Gregory Longmore Loza, A. J., Koride, S., Schimizzi, G. v., Li, B., Sun, S. X., & *Loza et al., 2016*. Cell density and actomyosin contractility control the organization of migrating collectives within an epithelium. *Molecular Biology of the Cell*, *27*(22), 3459–3,470. https://doi.org/10.1091/mbc.e16-05-0329 | N/A | N/A |
| Cell line (*Canis familiaris*) | MDCK I kidney, epithelial | ECACC | Cat. #: 00062106 | N/A |
| Cell line (*Canis familiaris*) | MDCK II kidney, epithelial | WashU Tissue Culture Center | N/A | N/A |
| Commercial assay or kit | RNeasy Mini Kit | Qiagen | Cat. #: 74106 | N/A |
| Commercial assay or kit | High-Capacity RNA-to-cDNA Kit | Thermo Fisher Scientific | Cat. #: 4368814 | N/A |

*Continued on next page*

*Continued*

| Reagent type (species) or resource | Designation | Source or reference | Identifiers | Additional information |
|---|---|---|---|---|
| Commercial assay or kit | TaqMan Fast Advanced Master Mix | Thermo Fisher Scientific | Cat. #: 4444556 | N/A |
| Sequence-based reagent | PTEN Hs02621230-s1 | Thermo Fisher Scientific | PCR primers | N/A |
| Sequence-based reagent | ZEB1 Hs00232783-m1 | Thermo Fisher Scientific | PCR primers | N/A |
| Sequence-based reagent | SNAIL1 Hs00195591-m1 | Thermo Fisher Scientific | PCR primers | N/A |
| Sequence-based reagent | TWIST1 Hs01675818-s1 | Thermo Fisher Scientific | PCR primers | N/A |
| Sequence-based reagent | GAPDH Hs02758991-g1 | Thermo Fisher Scientific | PCR primers | N/A |
| Sequence-based reagent | B2M Hs00187842-m1 | Thermo Fisher Scientific | PCR primers | N/A |
| Antibody | Anti-IκBα, mouse, monoclonal | Cell Signaling Technology | Cat. #: L35A5 | IF (1:400) |
| Antibody | Anti-catenin delta 1 (p120), rabbit, monoclonal | Cell Signaling Technology | Cat. #: D7S2M | IF (1:800) |
| Antibody | Anti-YAP, rabbit, monoclonal | Cell Signaling Technology | Cat. #: D8H1X | IF (1:100) |
| Antibody | Anti-P Myosin Light Chain 2, mouse, monoclonal | Cell Signaling Technology | Cat. #: 519 | IF (1:200) |
| Antibody | Anti-GM130, rabbit, monoclonal | Cell Signaling Technology | Cat. #: D6B1 | IF (1:200) |
| Antibody | Anti-Vimentin, rabbit, monoclonal | Cell Signaling Technology | Cat. #: 5741 S | IF: (1:100) |
| Antibody | Anti-ZO-1, mouse, monoclonal | Thermo Fisher Scientific | Cat. #: 33-9100 | IF: (1:100) |
| Antibody | AlexaFluor 488, goat anti-mouse, secondary | Life Technologies | Cat. #: A11001 | IF (1:500); secondary for IκBα, pMLC, ZO-1 |
| Antibody | AlexaFluor 488, goat anti-rabbit, secondary | Life Technologies | Cat. #: A11008 | IF (1:500); secondary for YAP |
| Antibody | AlexaFluor 647, goat anti-rabbit, secondary | Invitrogen | Cat. #: A31634 | IF (1:500); secondary for p120, GM130, vimentin |
| Chemical compound, drug | Rhodamine Phalloidin | Invitrogen | Cat. #: R415 | (1:250) |
| Chemical compound, drug | 40% Acrylamide Solution | BIORAD | Cat. #:1610140 | N/A |
| Chemical compound, drug | 2% Bis Solution | BIORAD | Cat. #: 1610142 | N/A |
| Chemical compound, drug | Bind silane | GE Healthcare Life Sciences | Cat. #: 17-1330-01 | N/A |
| Chemical compound, drug | Sigmacote | Sigma | Cat. #: SL2 | N/A |

*Continued on next page*

*Continued*

| Reagent type (species) or resource | Designation | Source or reference | Identifiers | Additional information |
|---|---|---|---|---|
| Chemical compound, drug | Ammonium persulfate | BIORAD | Cat. #: 1610700 | N/A |
| Chemical compound, drug | TEMED | Sigma | Cat. #: T7024 | N/A |
| Chemical compound, drug | HEPES sodium salt solution | Sigma | Cat. #: H3662 | N/A |
| Chemical compound, drug | Sulfo-SANPAH | Fisher Scientific, Proteochem | Cat. #: NC1314883 | N/A |
| Chemical compound, drug | Collagen I, rat tail | ChemCruz | Cat. #: sc-136157 | N/A |
| Chemical compound, drug | Leptomycin B | ChemCruz | Cat. #: sc-358688 | N/A |
| Chemical compound, drug | Selinexor | Selleck Chemicals | Cat. #: S7252 | N/A |
| Chemical compound, drug | Triton X-100 | Sigma | Cat. #: X100 | N/A |
| Chemical compound, drug | Bovine serum albumin | Sigma | Cat. #: A7906 | N/A |

## Polyacrylamide gel synthesis and collagen coating

Polyacrylamide gels were synthesized via free-radical polymerization, according to established protocols (*Fischer et al., 2012*). Precursor solutions were formulated from acrylamide, bis-acrylamide, and ultrapure water. To yield gel stiffnesses of 0.5, 2, and 50 kPa, respective acrylamide and bis-acrylamide percentages of 4%/0.2%, 5%/0.228%, and 12%/0.6% were used. Gels were attached to glass coverslips for immunofluorescent analyses and to glass-bottomed plates for time-lapse imaging. To facilitate gel attachment, these glass surfaces were activated by treatment with plasma and bind silane solution (94.7% ethanol, 5% acetic acid, and 0.3% bind silane for 10 min). After treatment, surfaces were rinsed with ethanol and air-dried. Precursor polyacrylamide solutions were polymerized by adding 10% ammonium sulfate and $N,N,N',N'$-tetramethylethylene (TEMED) at respective ratios of 1:200 and 1:2000 vol/vol. For immunofluorescent experiments, these final solutions were immediately dispensed onto Sigmacote-treated microscope slides and covered with plasma-activated glass coverslips; meanwhile, for time-lapse experiments, final solutions were dispensed onto plasma-activated glass-bottomed plates and covered with Sigmacote-treated glass coverslips. Solutions were allowed to polymerize for 30 min, then Sigmacote-treated surfaces were removed to expose one face of the formed gels. Gels were functionalized by 10-min incubation with Sulfo-SANPAH solution (50 mM HEPES (4-(2-hydroxyethyl)-1-piperazineethanesulfonic acid) buffer and 0.5 mg/ml Sulfo-SANPAH in $dH_2O$) under 365 nm UV. After rinsing, collagen type I solution (0.5 mg/ml in phosphate-buffered saline [PBS]) was applied, and gels were stored at 4°C overnight.

## Cell culture and colony seeding

All cell lines were maintained at 37°C and 5% $CO_2$. MCF10A cell lines were cultured in Dulbecco's Modified Eagle Medium (DMEM)/F12 1:1 supplemented with 5% horse serum, 20 ng/ml epidermal growth factor, 0.5 mg/ml hydrocortisone, 100 ng/ml cholera toxin, 10 µg/ml insulin, and 0.2% Normocin. Culture media for MDCK I cells consisted of MEM supplemented with Earle's salts, 2 mM glutamine, 10% fetal bovine serum (FBS), and 0.2% Normocin. Meanwhile, culture media for MDCK II cells consisted of DMEM supplemented with 10% FBS and 0.2% Normocin. Media was changed every 3 days during cell expansion. Cell line identities were authenticated using STR profiling, and cell lines tested negative for mycoplasma contamination. To prepare for cell colony seeding, collagen-coated polyacrylamide gels were rinsed gently with PBS and allowed to dry for 20 min. A volume of 5 µl, containing 18,000 cells, was then seeded at the center of each gel. Seeded cells were incubated for 24 hr to facilitate attachment and acclimatization to gel stiffness. After this incubation period, cells

were treated with 100 nM LMB (constituted in 100% ethanol), 500 nM Selinexor (SEL, constituted in 100% ethanol), or vehicle. Treated cell colonies were then used for immunofluorescent staining or time-lapse imaging.

## Immunofluorescence and confocal microscopy

Treated cells were cultured for 24 hr, then were fixed with 4% paraformaldehyde for 15 min. Fixed cells were rinsed and stored at 4°C until immunostaining. For cell membrane permeabilization, wells were incubated with 0.3% Triton X-100 in 2% wt/vol bovine serum albumin (BSA) at room temperature for 10 min. Then wells were blocked using 2% BSA in PBS, with overnight incubation at 4°C. Primary antibodies were incubated overnight at 4°C. After rinsing, secondary antibodies were also incubated overnight at 4°C. After secondary incubation, wells were rinsed and incubated with DAPI and phalloidin (if applicable) for 45 min. After final rinses, gels were coverslipped with mounting medium, and allowed to dry overnight. Immunostained cells were imaged at ×40 on a confocal microscope. To facilitate sample comparability, for all MCF10A ± LMB experiments, optimal capture settings were determined for each antigen based on the highest observed signals, then those settings were maintained for the duration of imaging. Imaging settings were changed to accommodate different signal intensities for MDCK cell lines and SINE experiments.

## Time-lapse microscopy

Time-lapse imaging was performed using a Zeiss AxioObserver Z1 microscope. Phase contrast images were acquired at 10-min intervals for a 24-hr period using a ×10 objective. For the duration of imaging, cell colonies were maintained at 37°C and 5% $CO_2$ using an attached incubation chamber.

## Quantitative image analysis

Unless otherwise stated, results for each condition (vehicle, LMB) reflect 8 FOVs from 2 biological replicates. Collected phase contrast time-lapse images were analyzed for the presence of multicellular streams. Inclusion criteria for streams were: (1) protrusion of 2 or more cells in succession from the monolayer baseline and (2) a protrusion length greater than twice its width (*Sarker et al., 2019*). To account for differences in image orientation, the number of streams counted over 24 hr was normalized to the baseline length of the migrating cell colony. All protein expression was measured in ImageJ. Nucleocytoplasmic (N:C) ratio was calculated as the nuclear intensity integrated over the nuclear area divided by the cytoplasmic intensity integrated over the cytoplasmic area. For YAP, N:C ratio was measured in at least 55 leader cells and 220 follower cells per condition. p120 expression ($n > 39$ leader cells and 55 follower cells from at least 6 FOVs reflecting 2 biological replicates per condition) was quantified as the average signal intensity integrated over a 10-μm line, where the line lay perpendicular to the junction and its midpoint rested at the junction center. Three measurements per cell–cell junction were averaged to define a single junction intensity. For α-cat KD, adhesions in cells treated with vehicle were mostly undetectable, so p120 expression was considered an unreliable measure. Instead, differences between groups were assessed via the number of visible adhesions. To measure ZO-1 expression, branch lengths were computed using the AnalyzeSkeleton plugin, where *n* reflected the number of branch measurements. Vimentin expression was measured as the integrated density within the monolayer region and normalized by the number of cells within the same region ($n = 10$ images per condition). pMLC and gm130 expression were measured as intensity within the monolayer, and were normalized to DAPI signal. Actin coherency was measured using the OrientationJ plugin, where *n* reflected the number of coherency measurements. Protrusions from the monolayer baseline were outlined manually, and the vector field was extracting using a grid size of 10 and an $\alpha$ value of 2. Because α-cat KD cells segregated into distinct colonies without clear polarization, coherency was measured for all cells. To assess Golgi orientation, the angle between the direction of migration and the line connecting a particular cell's nuclear centroid to its Golgi centroid was measured ($n > 50$ leader cells and 250 follower cells from at least 8 FOVs reflecting 2 biological replicates per condition). A polarized Golgi was considered to be one where this angle was less than 60°, as defined previously (*Mason et al., 2019*).

## Particle image velocimetry

Particle image velocimetry (PIV) was used to extract spatiotemporal velocity profiles from time-lapse images of migrating cell colonies. This analysis was performed using the MATLAB PIVlab package. The velocity field was calculated using 3 passes, with decreasing window sizes of 64, 32, and 16 pixels. All migration characteristics and heat maps were computed using MATLAB code ($n > 6$ FOV from 2 biological replicates per condition). The degree of cell cohesion was measured using order parameter, defined as the cosine of the angle between a single velocity vector and the average velocity vector for the respective time point. Meanwhile, when the value of that angle was greater or less than 90°, the vector was counted as backtracking.

## Traction force microscopy

To measure gel deformations, 1 μm fluorescent polystyrene beads were mixed with 0.5 kPa poly-acrylamide solution at a concentration of 20 million beads per mL gel solution. Images of beads at the gel surface (<10 μm depth) were captured during migration with vehicle or LMB. Three separate gels were analyzed per condition, with a total of 11 and 15 FOVs for vehicle and LMB-treated cells, respectively. An image of the relaxed beads, taken after cells were detached from gels using 0.5% trypsin, was used for reference to determine bead displacements. Bead displacements and cell-generated tractions were calculated using the Particle Image Velocimetry and Traction Force Microscopy ImageJ plugins, as described previously (*Tseng et al., 2012*). A minimum threshold of 2 Pa was set to remove potential noise from the data. Heat maps were generated using MATLAB code.

## Reverse transcription quantitative polymerase chain reaction

The expression of EMT marker genes was measured using reverse transcription quantitative polymerase chain reaction. RNA was extracted using the RNeasy Mini Kit (Qiagen), according to kit instructions. The extracted RNA was quantified using a NanoDrop Microvolume Spectrophotometer and was subsequently standardized to 2 ng/μl in nuclease free water. RNA was converted to cDNA using the High-Capacity RNA-to-cDNA Kit (Applied Biosystems), according to kit instructions. qPCR was performed using TaqMan Fast Advanced Master Mix (Applied Biosystems) according to product instructions, with probes for PTEN, ZEB1, SNAIL1, TWIST1, GAPDH, and B2M. EMT genes were normalized to the GAPDH housekeeping gene to compare relative expression levels between groups ($n = 12$ wells from 4 biological replicates per condition).

## Statistical analysis

Bar and line plots are presented as the mean ± standard error of the mean. Statistical significance was determined using two-way analysis of variance with Tukey post hoc comparisons. For tractions and all measurements from MDCK I, II, and Selinexor experiments, significance was assessed using two-tailed *t*-test. Differences were considered significant for *$p < 0.05$, **$p < 0.01$, ***$p < 0.001$, ****$p < 0.0001$.

## Acknowledgements

This work was supported by the NIH/NIGMS MIRA (R35GM128764) grant to AP.

## Additional information

### Funding

| Funder | Grant reference number | Author |
| --- | --- | --- |
| National Institutes of Health | R35GM128764 | Amit Pathak |

The funders had no role in study design, data collection, and interpretation, or the decision to submit the work for publication.

## Author contributions
Carly M Krull, Conceptualization, Resources, Data curation, Software, Formal analysis, Validation, Investigation, Visualization, Methodology, Writing – original draft, Writing – review and editing; Haiyi Li, Validation, Methodology; Amit Pathak, Conceptualization, Supervision, Funding acquisition, Investigation, Methodology, Project administration, Writing – review and editing

## Author ORCIDs
Carly M Krull ![ORCID] http://orcid.org/0000-0002-8170-4282
Amit Pathak ![ORCID] http://orcid.org/0000-0003-4006-5119

## Decision letter and Author response
Decision letter https://doi.org/10.7554/eLife.81048.sa1
Author response https://doi.org/10.7554/eLife.81048.sa2

## Additional files

### Supplementary files
• MDAR checklist

• Source code 1. Code for extracting migration parameters from timelapse and particle image velocimetry data.

### Data availability
Source codes used for data analysis and plotting figures are provided as a zip supplementary file.

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
