## [Editor Report]

This work is an important contribution to our understanding of epithelial migration. Previous work had shown that nuclear export inhibition (NEI), which is employed as a therapeutic strategy to treat cancer, traps several known regulators of epithelial-mesenchymal (E-M) phenotypes; however, how NEI alters the mechano-response and collective cell migration of healthy epithelia on substrates of varying stiffness was not described. The convincing new results show that NEI induces an intermediate E-M state where cells concurrently strengthen intercellular adhesions and develop mechano-active characteristics. Migration of epithelial monolayers becomes disordered and leads to multicellular streaming.

---

## [Decision Letter]

**Decision letter after peer review:**

Thank you for submitting your article "Nuclear export inhibition jumbles epithelial-mesenchymal states and gives rise to migratory disorder in healthy epithelia" for consideration by *eLife*. Your article has been reviewed by 3 peer reviewers, and the evaluation has been overseen by a Reviewing Editor and Jonathan Cooper as the Senior Editor. The following individual involved in review of your submission has agreed to reveal their identity: Elsa Bazellières (Reviewer #1).

Essential revisions:

The Reviewers all agree that the central message of the manuscript is of strong potential interest to the community. On the other hand, they also agree that a number of additional experiments are needed to confirm this result and to consolidate the conclusions of the paper. Their comments should allow you to prepare a more complete revised version with additional experiments that would significantly improve the value and impact of this work.

Please consider the following points in particular:

1) The reviewers agree that it is unfortunate that the data are based on a single cell line and are not validated by other cell models. We would like to have confirmation that the results of your paper do not apply only to a particular cell line.

2) Your work relies heavily on a single nuclear protein export inhibitor, leptomycin B, but we all agree with Reviewer 2 that additional positive controls are needed to strengthen the results.

3) Reviewer 1 also suggests the use of additional standard EMT markers to better characterize the state of the cells, which is a good idea.

*Reviewer #1 (Recommendations for the authors):*

This manuscript could be of interest to the broad community of scientists working on collective cell migration and tumor processes. However, this work has many shortcomings that need to be addressed.

In this study, the work focuses on the impact of nuclear export inhibition (NEI) during the complex process of epithelial to mesenchymal transition. Over the past decade, several studies have demonstrated that CRM1-related NEI is capable of reversing the EMT process. Inhibition of CRM1 has been developed as a therapeutic strategy against cancer, as it has been shown that inhibition of the nuclear export receptor CRM1 is able to return oncogenic and tumor suppressor proteins to the nucleus, thereby restoring cell cycle regulation. However, the underlying molecular and cellular mechanisms are still difficult to understand. To address this issue, the authors use MCF10A cells with the CRM1 inhibitor Leptomycin B (LMB), and gels with different stiffnesses to completely inhibit nuclear export or to modulate nucleocytoplasmic transport dynamics respectively. The attractive conclusion is that NEI forces competition between cellular E-M states, which leads to disorded collective cell migration.

The proposed study by Krull et al. is interesting, but I am not convinced that the results presented have the potential significance, no mechanistic explanation and weak description of cellular features, with no good makers for epithelail or mesenchymal state, and robustness required. Many experimental aspects are not precise and therefore it is difficult to rely on the data presented (e.g. stainings are not convincing, a lack of explanation for quantifications, only 2 biological replicates, only one cell line…etc). In the current state of the study, we cannot raise the conclusion made in panel 1 event if it could be attractive.

The manuscript lacks also a second cell type, and an appropriate methodology to study EMT. The authors should also defend the choice of MCF10A as a model system, as these cells do not show clear EMT characteristics. The authors state otherwise but no reference is given (line 101). Alternatively, the authors may choose to focus on collective cell migration without emphasizing the EMT process. MCF10A seeding protocol: no EMT process, cells are seeded in small colonies and thus migrate directly. The set up of the experiment is in itself not adapted to the study of EMT.

Comments on the manuscript:

Figure 2:

The authors did not comment on the impact of gel stiffnesses on cell and nuclear morphology. In the images, the density of cells on the 2kPa gel is low compared to the other conditions. MCF10A cells appear larger when seeded on 2kPa gels and the nuclei are larger, possibly because the cells are flatter. The opposite phenotype is observed when MCF10A cells are seeded on 50kPa gels (ETOH condition). Nuclear and cell morphology, but also cell density may have an impact on NC transport and should be described and quantified.

Comparisons between the different conditions could be biased if the seeding density is not correct on the 2kPa gels. In order to avoid biased and unreliable conclusions, this point needs to be clarified.

High lkBa level is detected in the cytoplasm of 0.5KPa gels and even more present in +LMB conditions, with less recruitment around cell-cell junctions. Can the authors comment on the fact that simply placing the cells in different rigidities affects the overall level of lkBa expression? How exactly did the authors quantify the ratio of LkBa? In the methods, it is not clear whether they measured the intensity per single cell or over the entire fields of view without segmenting each cell. The ratio should also be shown in the leader versus followers.

Figure 3:

YAP panel: According to the literature, YAP is supposed to be translocated into nuclei on high stiffness gels (30kPa, Elosegui Artola, Cell 2017), which does not seem to be the case in Figure 3 panel B ETOH conditions. In the quantification, YAP seems to be more localized in the cytoplasm as the values are always lower than 1. Can the authors comment on this discrepancy between their data and the literature?

In the text, the authors mention the leader cell and the follower cell, but the data are not presented. They should add the corresponding graphs.

pMLC panel: the color of the scale bar should be changed, in order to see the different low intensities in the images. As it is, image interpretation is impossible and the lack of DAPI staining makes quantification unreliable. I am not entirely convinced of the need to normalize the intensity of pMLC to DAPI, what is the reason to do this?

Only in the 50kPa condition, we see a drastic difference in pMLC that is not clear in the graphs. How do the authors take into account the differences in the +ETOH conditions to conclude the effect of NEI?

In the text, line 184, do the higher cytoskeletal forces and front rear polarization refer to the organization of pMLC and actin respectively? If so, the sentence should be reworded to be clearer and the pMLC and actin channels should be merged into an additional panel in B to demonstrate colocalization of pMLC and actin fibers. To do this, the image quality should be increased and a higher magnification 63X objective with good NA should be used.

Figure 4:

In the text, the title of the paragraph should be changed because only velocity is shown, no data on cell adhesion enhancement is presented in the figure or supplementary data.

The conclusion should be reworded because the increased migration is only observed during the first few hours of experimentation. The link to Figure 5 is not clear in the text.

Figure 5:

There is no experimental evidence at all that gm130 expression supports cytoskeletal rearrangement, like many of the findings in this study, this is overstated. Furthermore, loss of polarization may influence how cells migrate without affecting speed, the correlation is unclear.

In 2kPa gels, cell density seems to be decreased and this could obviously affect cell behavior.

Quantification of gm130 expression based on the intensity ratio between gm130 and DAPI is not a robust way to quantify protein expression. A western blot should be performed.

Figure 7:

The same comments as above can be made for this figure, so please report the comments for Figures 2, 3 and 4.

The control panels are missing in A, D, H, J, K, L, and M, making this figure unreliable. Can the authors show the raw data for the graphs as it appears that the values in the WT are the same as in the previous figure. An additional check with shYAP scrambling should be performed to avoid bias.

The punctate pattern of LkBa is not visible. To demonstrate that the protein is degraded, the degradation pathways must be blocked and a reversion of the phenotype must be obtained. The punctate motif can also be obtained by disrupting the anchoring of the protein, as YAP can affect the expression of many proteins.

Figure 8:

The same comments as above can be made for this figure, so please report the comments for Figures 2, 3 and 4.

Figure 9:

A: To assert that there is competition between epithelial and mesenchymal features, much more precise and accurate analyses and experiments are needed. The authors should compare the localization of YAP in streams with the localization of YAP outside of streams. More epithelial markers are needed, such as Ecadherin staining, tight junction proteins, vinculin, laser ablation experiments…, and differences should be observed between cells in the streams and cells outside.

In the current state of the study, we cannot raise the conclusion made in panel 1 event if it might be attractive.

*Reviewer #2 (Recommendations for the authors):*

In this article Krull and colleagues evaluate the impact of nuclear export inhibition on epithelial to mesenchymal transition. They use a well recognized nuclear transport inhibitor Leptomycin B (LMB) to reveal two distinct set of markers being regulated as a consequence of nuclear export inhibition (NEI) (a) soft substrates elevate collective migration and (b) while stiffer substrates reduce migration at all time points. Mechanistically, the authors focus on Yes associated protein YAP and α catenin. Knockdown studies reveal that the former could shifts affected cells toward an epithelial phenotype while the latter maintained intercellular adhesion.

Strengths of the paper include the parsing of two distinct subsets of targets for their role in morphology transition states upon NEI. Strengths also include detailed mechanistic studies deciphering the role of YAP and α catenin protein in the distinct mechano regulatory pathways during cell state transition.

In terms of weakness, the authors rely heavily on one nuclear protein export inhibitor leptomycin B. Additional positive controls are needed to strengthen the results.

Overall the results (even though with one inhibitor) are supportive of the hypothesis and are clearly presented.

The results provide a shift in our understanding of EMT which may not be in continuum rather, opposing transcriptional programs can collectively give rise to epithelial and highly mesenchymal characteristics that is being presented as a concurrent transition of epithelial to mesenchymal state.

Specific comments:

Figure 1 is redundant and can be merged elsewhere.

Positive controls like selinexor, eltanexor or other XPO1/CRM1 inhibitors should be included in some of the experiments to support the overall conclusions.

Chemical inhibition results through LMB are really strong. However, were similar results obtained by biological inactivation of XPO1? What is the consequence on EMT state (both soft substrates and stiffer substrates) on XPO1 si/shRNA knockdown?

Leptomycin is a irreversible inhibitor while SINE compounds are slowly reversible inhibitors of CRM1/XPO1. Studies should check whether mode of NEI (permanent vs transient) impacts differently the transient process of E to M state transition.

*Reviewer #3 (Recommendations for the authors):*

In this manuscript, the authors have demonstrated that NEI maintains an atypical E-M state where it strengthens intercellular adhesions and develops mechanoactivation simultaneously. NEI augments collective migration only on soft substrates as opposed to stiffer substrates where the migration is reduced. Furthermore, the authors have shown that YAP1 depletion from NEI positions cells toward an epithelial phenotype whereas knockdown of α-catenin shifts the cells toward a mesenchymal state. Overall, the manuscript is (mostly) well written and their claims are well supported by their data. The data, however, relies on just one cell line and lacks validation in other cell-based models. The study would contribute significantly to basic and translational research.

---

## [Author Response]

Essential revisions:The Reviewers all agree that the central message of the manuscript is of strong potential interest to the community. On the other hand, they also agree that a number of additional experiments are needed to confirm this result and to consolidate the conclusions of the paper. Their comments should allow you to prepare a more complete revised version with additional experiments that would significantly improve the value and impact of this work.Please consider the following points in particular:1) The reviewers agree that it is unfortunate that the data are based on a single cell line and are not validated by other cell models. We would like to have confirmation that the results of your paper do not apply only to a particular cell line.2) Your work relies heavily on a single nuclear protein export inhibitor, leptomycin B, but we all agree with Reviewer 2 that additional positive controls are needed to strengthen the results.3) Reviewer 1 also suggests the use of additional standard EMT markers to better characterize the state of the cells, which is a good idea.

We thank the editor and the reviewers for their positive comments about our work and providing thoughtful feedback. In this revision, we have addressed all reviewer concerns through several new experiments and revision of text, as necessary, all of which have improved the manuscript. With regards to essential revisions:

1) We have now added data for two additional epithelial cell lines, MDCK-I and MDCK-II, with and without LeptomycinB (LMB) treatment on soft (0.5kPa) hydrogels, which was the main control condition in original submission. Indeed, we found that nuclear export inhibition (NEI) alters their epithelial (E) and mesenchymal (M) states (Figure 3 —figure supplement 3). Specifically, NEI shifts both cell lines towards mesenchymal states while still maintaining multicellular structures in the form of collective streams. These additional results validate and strengthen our main finding that NEI jumbles E/M states of epithelial cells. We also discuss that the nature of such NEI-induced E/M changes is not expected to be identical across cell lines/types because they vary in their genetic and proteomic identities, especially in their epithelial/mesenchymal starting points. Inclusion of additional cell lines further shows that the effect of NEI on cells’ epithelial and mesenchymal states is a broad and robust response, which is currently underappreciated in literature.

2) We performed NEI using Selinexor on MCF10A cells on soft gels (control condition) and repeated our original measurements (Figure 3 —figure supplement 2). Indeed, Selinexor validates results from LMB treatment for otherwise same conditions (cell lines and hydrogel) and shows elevated epithelial and mesenchymal features.

3) We have now added qPCR and immunostaining (as applicable) measurements for additional epithelial markers (ZO-1 and PTEN) and mesenchymal markers (vimentin, ZEB1, TWIST1, SNAIL1). These measurements (revised Figure 2E-F, Figure 3A-E) confirm and solidify our original findings that both epithelial and mesenchymal markers were prominent after NEI.

Reviewer #1 (Recommendations for the authors):This manuscript could be of interest to the broad community of scientists working on collective cell migration and tumor processes. However, this work has many shortcomings that need to be addressed.In this study, the work focuses on the impact of nuclear export inhibition (NEI) during the complex process of epithelial to mesenchymal transition. Over the past decade, several studies have demonstrated that CRM1-related NEI is capable of reversing the EMT process. Inhibition of CRM1 has been developed as a therapeutic strategy against cancer, as it has been shown that inhibition of the nuclear export receptor CRM1 is able to return oncogenic and tumor suppressor proteins to the nucleus, thereby restoring cell cycle regulation. However, the underlying molecular and cellular mechanisms are still difficult to understand. To address this issue, the authors use MCF10A cells with the CRM1 inhibitor Leptomycin B (LMB), and gels with different stiffnesses to completely inhibit nuclear export or to modulate nucleocytoplasmic transport dynamics respectively. The attractive conclusion is that NEI forces competition between cellular E-M states, which leads to disorded collective cell migration.The proposed study by Krull et al. is interesting, but I am not convinced that the results presented have the potential significance, no mechanistic explanation and weak description of cellular features, with no good makers for epithelail or mesenchymal state, and robustness required. Many experimental aspects are not precise and therefore it is difficult to rely on the data presented (e.g. stainings are not convincing, a lack of explanation for quantifications, only 2 biological replicates, only one cell line…etc). In the current state of the study, we cannot raise the conclusion made in panel 1 event if it could be attractive.The manuscript lacks also a second cell type, and an appropriate methodology to study EMT. The authors should also defend the choice of MCF10A as a model system, as these cells do not show clear EMT characteristics. The authors state otherwise but no reference is given (line 101). Alternatively, the authors may choose to focus on collective cell migration without emphasizing the EMT process. MCF10A seeding protocol: no EMT process, cells are seeded in small colonies and thus migrate directly. The set up of the experiment is in itself not adapted to the study of EMT.

We thank the reviewer for recognizing and listing strengths of our work and for providing a detailed review of the manuscript. In this revision, we have addressed all concerns through a variety of new experiments, analyses, explanations, and revised text. We have now added two cell lines (MDCK-I and MDCK-II) to show again that NEI alters E-M states. We have added mechanistic explanations in the revised Discussion section. Our cell seeding protocol is similar to many previous EMT studies, at least the ones that use hydrogels of tunable stiffness and not the unrealistically stiff tissue culture plastic (Lee *et al.*, 2012; Leight *et al.*, 2012; Brown *et al.*, 2013; Wei *et al.*, 2015; Nasrollahi and Pathak, 2016; Matte *et al.*, 2018; Walter *et al.*, 2018; Sarker *et al.*, 2019). In these studies, and in ours, epithelial clusters or colonies of consistent density are seeded and EMT markers are measured. We appreciate the reviewer point around cell density and its potential effect, which we address in this letter, as described below. We have also added several new EMT markers (ZO-1, PTEN, vimentin, ZEB1, TWIST1, SNAIL1) that show E/M changes according to cell/matrix/NEI conditions.

Comments on the manuscript:Figure 2:The authors did not comment on the impact of gel stiffnesses on cell and nuclear morphology. In the images, the density of cells on the 2kPa gel is low compared to the other conditions. MCF10A cells appear larger when seeded on 2kPa gels and the nuclei are larger, possibly because the cells are flatter. The opposite phenotype is observed when MCF10A cells are seeded on 50kPa gels (ETOH condition). Nuclear and cell morphology, but also cell density may have an impact on NC transport and should be described and quantified.Comparisons between the different conditions could be biased if the seeding density is not correct on the 2kPa gels. In order to avoid biased and unreliable conclusions, this point needs to be clarified.High lkBa level is detected in the cytoplasm of 0.5KPa gels and even more present in +LMB conditions, with less recruitment around cell-cell junctions. Can the authors comment on the fact that simply placing the cells in different rigidities affects the overall level of lkBa expression? How exactly did the authors quantify the ratio of LkBa? In the methods, it is not clear whether they measured the intensity per single cell or over the entire fields of view without segmenting each cell. The ratio should also be shown in the leader versus followers.

We considered the reviewer’s point about cell density, further inspected our fixed images, and found that any apparent differences between images do not appear to be trends across the stiffnesses. As shown in Author response image 1, IκBα N:C ratio was consistent across stiffness for control cells. Cells treated with LMB trended toward higher cell density on 0.5 kPa, but there were no other clear differences. Importantly, we made sure to seed the same number of cells from the same vial of cells in every gel sample, as described in Methods. Thus, any potential density effects are automatically normalized across conditions.

**Author response image 1. sa2fig1:** 

Figure 3:YAP panel: According to the literature, YAP is supposed to be translocated into nuclei on high stiffness gels (30kPa, Elosegui Artola, Cell 2017), which does not seem to be the case in Figure 3 panel B ETOH conditions. In the quantification, YAP seems to be more localized in the cytoplasm as the values are always lower than 1. Can the authors comment on this discrepancy between their data and the literature?In the text, the authors mention the leader cell and the follower cell, but the data are not presented. They should add the corresponding graphs.pMLC panel: the color of the scale bar should be changed, in order to see the different low intensities in the images. As it is, image interpretation is impossible and the lack of DAPI staining makes quantification unreliable. I am not entirely convinced of the need to normalize the intensity of pMLC to DAPI, what is the reason to do this?Only in the 50kPa condition, we see a drastic difference in pMLC that is not clear in the graphs. How do the authors take into account the differences in the +ETOH conditions to conclude the effect of NEI?In the text, line 184, do the higher cytoskeletal forces and front rear polarization refer to the organization of pMLC and actin respectively? If so, the sentence should be reworded to be clearer and the pMLC and actin channels should be merged into an additional panel in B to demonstrate colocalization of pMLC and actin fibers. To do this, the image quality should be increased and a higher magnification 63X objective with good NA should be used.

We have checked and modified colorbars for images (actin, pMLC) that may have been unclear in original submission. Any interpretations around actin and pMLC are separate points and thus co-localization would not provide additional insights, in our opinion, especially since merging of channels was not possible in all cases. We have corrected text around leader and follower cells, which was a qualitative observation. We decided not to quantify ‘follower’ cells because those vary quite a bit according to distance from the leading edge. Since leader cells are easier to define and distinguish (first row of cells touching the leading edge), we have included leader cell analyses wherever it was applicable. For YAP quantification and comparison, we have noted in Methods:

“Nucleocytoplasmic (N:C) ratio was calculated as the nuclear intensity integrated over the nuclear area divided by the cytoplasmic intensity integrated over the cytoplasmic area. Thus, unlike Elosegui Artola, Cell 2017, our calculation accounts for area, which explains values less than 1. The main purpose of this calculation was to compare WT against LMB and the same method was used for both, so any differences merely point to higher nuclear localization after LMB, which is also clear from images. Regarding the actin/pMLC imaging – since we are using hydrogels of varying stiffness, we are limited by the objective magnification. We tried 63x earlier but imaging was not as consistent as it is with 40x. Since hydrogels of different stiffnesses have different refractive indices (stiffer gels pass less light), pixel intensity can change. We kept the same laser intensity across samples for sake of uniform exposure. Thus, to account for potential changes in refractive index of gels, normalizing by DAPI in each image was a way to make consistent quantifications.”

Figure 4:In the text, the title of the paragraph should be changed because only velocity is shown, no data on cell adhesion enhancement is presented in the figure or supplementary data.The conclusion should be reworded because the increased migration is only observed during the first few hours of experimentation. The link to Figure 5 is not clear in the text.

We have modified the title of the paragraph. We have now clarified that negative velocities refer to velocity vectors in directions away from leading-edge advancement. We have also ensured that methods describe velocity calculation. Since actin fiber orientation came from fixed imaging, it cannot be directly and legitimately correlated with collective cell migration order. We make a note of that.

Figure 5:There is no experimental evidence at all that gm130 expression supports cytoskeletal rearrangement, like many of the findings in this study, this is overstated. Furthermore, loss of polarization may influence how cells migrate without affecting speed, the correlation is unclear.In 2kPa gels, cell density seems to be decreased and this could obviously affect cell behavior.Quantification of gm130 expression based on the intensity ratio between gm130 and DAPI is not a robust way to quantify protein expression. A western blot should be performed.

We are in agreement with the reviewer that disorder in polarization alters how cells migrate without changing speed, which was precisely our intention behind calculating backtracking cells and focusing on ‘disorder’ due to NEI as opposed to just speed (as majority of previous collective cell migration studies have done). Indeed, for this reason we emphasized “migratory disorder” in the title of the manuscript.

We note that cytoskeleton rearrangement and golgi reorientation are both required if cells need to repolarize, which is the rationale behind using golgi imaging. For this very reason, many previous studies have imaged for golgi using gm130 and measured its polarization relative to cells’ front-rear polarity in collective and individual cells, without really performing western blot because net protein expression was not the main emphasis (Matsuzawa *et al.*, 2018; Mason *et al.*, 2019; Hino *et al.*, 2020; Rong *et al.*, 2021). We agree that immunostaining is not the best way to quantify gm130 expression, which was not the primary goal here. Instead, our intention with gm130 imaging was to quantify cell polarity, as previously done, which is not possible using a western blot. This also connects to our point about ‘disorder’, which reviewer also appreciated. After analyzing golgi polarization from gm130 images, we further used these immune-stained images to analyze gm130 expression and presented the somewhat surprising finding of increased gm130 expression that also supports increased vimentin expression after LMB.

Figure 7:The same comments as above can be made for this figure, so please report the comments for Figures 2, 3 and 4.The control panels are missing in A, D, H, J, K, L, and M, making this figure unreliable. Can the authors show the raw data for the graphs as it appears that the values in the WT are the same as in the previous figure. An additional check with shYAP scrambling should be performed to avoid bias.The punctate pattern of LkBa is not visible. To demonstrate that the protein is degraded, the degradation pathways must be blocked and a reversion of the phenotype must be obtained. The punctate motif can also be obtained by disrupting the anchoring of the protein, as YAP can affect the expression of many proteins.Figure 8:The same comments as above can be made for this figure, so please report the comments for Figures 2, 3 and 4.

Since control panels have already been provided in earlier figures, we thought it would be repetitive to include those here again, which could also constitute as padding the figure with already used data. Moreover, this figure is already quite full and repeating those data would make it overcrowded and hard to understand, in our opinion. Here, shYAP and αCat-KD are implemented in MCF10A and can be compared against WT MCF10A cells. Although scramble could add value, we don’t believe it would add significant new insights and thus decided not to pursue this in revision. So, instead, we focused our limited time and resources on more “essential revisions” and performed experiments and analyses that have bolstered our key message of NEI-induced E/M alterations.

Figure 9:A: To assert that there is competition between epithelial and mesenchymal features, much more precise and accurate analyses and experiments are needed. The authors should compare the localization of YAP in streams with the localization of YAP outside of streams. More epithelial markers are needed, such as Ecadherin staining, tight junction proteins, vinculin, laser ablation experiments…, and differences should be observed between cells in the streams and cells outside.In the current state of the study, we cannot raise the conclusion made in panel 1 event if it might be attractive.

We respectfully note that our results with the loss of YAP and α-Catenin are indeed novel because no other study has reported what happens to EMT states on surfaces of varying stiffnesses after perturbations in these two proteins. Moreover, no other study has shown how these two proteins regulate NEI-induced E/M changes, which the reviewer has already noted. We have added numerous measurements of E/M markers, nucleocytoplasmic factors, and collective migration: ZO-1, PTEN, vimentin, ZEB1, TWIST1, SNAIL1, p120, IκΒα, YAP, Actin, pMLC, GM130, traction force microscopy, single cell and multicellular analyses of collectively migrating streams, 2 different NEI drugs, 5 different cell lines (including knockdowns), 3 different hydrogel stiffnesses, golgi polarization, and particle image velocimetry for temporal analyses of migration speed/order/backtracking. All measurements for these markers were indeed performed precisely and accurately to the best of our knowledge, as much as the existing methodology allows and consistent with many published studies.

Reviewer #2 (Recommendations for the authors):In this article Krull and colleagues evaluate the impact of nuclear export inhibition on epithelial to mesenchymal transition. They use a well recognized nuclear transport inhibitor Leptomycin B (LMB) to reveal two distinct set of markers being regulated as a consequence of nuclear export inhibition (NEI) (a) soft substrates elevate collective migration and (b) while stiffer substrates reduce migration at all time points. Mechanistically, the authors focus on Yes associated protein YAP and α catenin. Knockdown studies reveal that the former could shifts affected cells toward an epithelial phenotype while the latter maintained intercellular adhesion.Strengths of the paper include the parsing of two distinct subsets of targets for their role in morphology transition states upon NEI. Strengths also include detailed mechanistic studies deciphering the role of YAP and α catenin protein in the distinct mechano regulatory pathways during cell state transition.In terms of weakness, the authors rely heavily on one nuclear protein export inhibitor leptomycin B. Additional positive controls are needed to strengthen the results.Overall the results (even though with one inhibitor) are supportive of the hypothesis and are clearly presented.The results provide a shift in our understanding of EMT which may not be in continuum rather, opposing transcriptional programs can collectively give rise to epithelial and highly mesenchymal characteristics that is being presented as a concurrent transition of epithelial to mesenchymal state.

We thank the reviewer for this thoughtful summary of strengths of our paper and for providing valuable feedback. We agree that additional controls were needed, which we have included in this revision by validating key EMT phenotypes using another NEI drug.

Specific comments:Figure 1 is redundant and can be merged elsewhere.

We appreciate the reviewer’s suggestion, which we thoughtfully considered. However, considering the broad readership of *eLife*, we thought it would be better to keep this figure to introduce our hypothesis around nuclear export and visualize experimental design. We kept the schematic in Figure 2, which shows one already established mechanism of NEI, shown in cancer cells. That said, we have removed the schematic in Figure 3, which somewhat captures some of these ideas and its placement in Figure 1 would be better for readability.

Positive controls like selinexor, eltanexor or other XPO1/CRM1 inhibitors should be included in some of the experiments to support the overall conclusions.

We thank the reviewer for this suggestion. We repeated experiments with Selinexor, another NEI drug, and found similar elevation of both epithelial and mesenchymal traits, which is included in the revised manuscript.

Chemical inhibition results through LMB are really strong. However, were similar results obtained by biological inactivation of XPO1? What is the consequence on EMT state (both soft substrates and stiffer substrates) on XPO1 si/shRNA knockdown?

Unlike LMB and Selinexor, biological inactivation of XPO1 would likely be much more cytotoxic and would complicate measurements of live cell migration and EMT. Thus, we instead relied on a given NEI method and instead focused on studying its functional outcomes in mechanosensitve collective cell migration and E-M states.

Leptomycin is a irreversible inhibitor while SINE compounds are slowly reversible inhibitors of CRM1/XPO1. Studies should check whether mode of NEI (permanent vs transient) impacts differently the transient process of E to M state transition.

We have now included results after treatment with Selinexor, which is a reversible inhibitor nuclear export.

Reviewer #3 (Recommendations for the authors):In this manuscript, the authors have demonstrated that NEI maintains an atypical E-M state where it strengthens intercellular adhesions and develops mechanoactivation simultaneously. NEI augments collective migration only on soft substrates as opposed to stiffer substrates where the migration is reduced. Furthermore, the authors have shown that YAP1 depletion from NEI positions cells toward an epithelial phenotype whereas knockdown of α-catenin shifts the cells toward a mesenchymal state. Overall, the manuscript is (mostly) well written and their claims are well supported by their data. The data, however, relies on just one cell line and lacks validation in other cell-based models. The study would contribute significantly to basic and translational research.

We thank the reviewer for positive comments about this work. We agree that some validation in another cell line is needed. In this revision, we have added two variants of a popular epithelial cell line – MDCK-I and MDCK-II. In both these cell lines, NEI jumbles E-M states, which is consistent with the key finding of our original submission.

References:

Brown, AC, Fiore, VF, Sulchek, TA, and Barker, TH (2013). Physical and chemical microenvironmental cues orthogonally control the degree and duration of fibrosis-associated epithelial-to-mesenchymal transitions. J Pathol 229, 25–35.

Hino, N, Rossetti, L, Marín-Llauradó, A, Aoki, K, Trepat, X, Matsuda, M, and Hirashima, T (2020). ERK-Mediated Mechanochemical Waves Direct Collective Cell Polarization. Dev Cell 53, 646-660.e8.

Lee, K, Chen, QK, Lui, C, Cichon, MA, Radisky, DC, and Nelson, CM (2012). Matrix compliance regulates Rac1b localization, NADPH oxidase assembly, and epithelial-mesenchymal transition. Mol Biol Cell 23, 4097–4108.

Leight, JL, Wozniak, MA, Chen, S, Lynch, ML, and Chen, CS (2012). Matrix rigidity regulates a switch between TGF-beta1-induced apoptosis and epithelial-mesenchymal transition. Mol Biol Cell 23, 781–791.

Mason, DE, Collins, JM, Dawahare, JH, Nguyen, TD, Lin, Y, Voytik-Harbin, SL, Zorlutuna, P, Yoder, MC, and Boerckel, JD (2019). YAP and TAZ limit cytoskeletal and focal adhesion maturation to enable persistent cell motility. J Cell Biol 46, jcb.201806065-21.

Matsuzawa, K, Himoto, T, Mochizuki, Y, and Ikenouchi, J (2018). α-Catenin Controls the Anisotropy of Force Distribution at Cell-Cell Junctions during Collective Cell Migration. Cell Rep 23, 3447–3456.

Matte, BF, Kumar, A, Placone, JK, Zanella, VG, Martins, MD, Engler, AJ, and Lamers, ML (2018). Matrix stiffness mechanically conditions EMT and migratory behavior of oral squamous cell carcinoma. J Cell Sci, jcs.224360.

Nasrollahi, S, and Pathak, A (2016). Topographic confinement of epithelial clusters induces epithelial-to-mesenchymal transition in compliant matrices. Sci Rep 6, 18831.

Rong, Y et al. (2021). The Golgi microtubules regulate single cell durotaxis. Embo Rep 22, e51094.

Sarker, B, Bagchi, A, Walter, C, Almeida, J, and Pathak, A (2019). Longer collagen fibers trigger multicellular streaming on soft substrates via enhanced forces and cell–cell cooperation. J Cell Sci 132, jcs226753.

Walter, C, Davis, JT, Mathur, J, and Pathak, A (2018). Physical defects in basement membrane-mimicking collagen-IV matrices trigger cellular EMT and invasion. Integr Biol (Camb), 1–14.

Wei, SC et al. (2015). Matrix stiffness drives epithelial-mesenchymal transition and tumour metastasis through a TWIST1-G3BP2 mechanotransduction pathway. Nature Cell Biology 17, 678–688.